# A nonenzymatic dependency on inositol-requiring enzyme 1 controls cancer cell cycle progression and tumor growth

Iratxe Zuazo-Gaztelu[1], David Lawrence[1], Ioanna Oikonomidi[1], Scot Marsters[1], Ximo Pechuan-Jorge[1], Catarina J. Gaspar[1], David Kan[2], Ehud Segal[2], Kevin Clark[3], Maureen Beresini[3], Marie-Gabrielle Braun[4], Joachim Rudolph[4], Zora Modrusan[5], Meena Choi[5], Wendy Sandoval[5], Mike Reichelt[6], David C. DeWitt[6], Pekka Kujala[7], Suzanne van Dijk[7], Judith Klumperman[7], Avi Ashkenazi[1]*

1 Department of Research Oncology, Genentech, Inc., South San Francisco, California, United States of America, 2 Department of In Vivo Pharmacology, Genentech, Inc., South San Francisco, United States of America, 3 Department of Biochemical and Cellular Pharmacology, Genentech, Inc., South San Francisco, California, United States of America, 4 Department of Discovery Chemistry, Genentech, Inc., South San Francisco, California, United States of America, 5 Department of Proteomic and Genomic Technologies, Genentech, Inc., South San Francisco, California, United States of America, 6 Department of Pathology, Genentech, Inc., South San Francisco, California, United States of America, 7 Center for Molecular Medicine—Cell Biology, University Medical Center, Utrecht, The Netherlands

* aa@gene.com

## Abstract

Endoplasmic-reticulum resident inositol-requiring enzyme 1α (IRE1) supports protein homeostasis via its cytoplasmic kinase-RNase module. Known cancer dependency on IRE1 entails its enzymatic activation of the transcription factor XBP1s and of regulated RNA decay. We discovered surprisingly that some cancer cell lines require IRE1 but not its enzymatic activity. IRE1 knockdown but not enzymatic IRE1 inhibition or XBP1 disruption attenuated cell cycle progression and tumor growth. IRE1 silencing led to activation of TP53 and CDKN1A/p21 in conjunction with increased DNA damage and chromosome instability, while decreasing heterochromatin as well as DNA and histone H3K9me3 methylation. Immunoelectron microscopy detected some endogenous IRE1 protein at the nuclear envelope. Thus, cancer cells co-opt IRE1 either enzymatically or nonenzymatically, which has significant implications for IRE1's biological role and therapeutic targeting.

## Introduction

Eukaryotic cells rely on a membrane-surrounded organelle known as the endoplasmic reticulum (ER) to fold newly synthesized transmembrane and secreted proteins. Inositol-requiring enzyme 1α (*ERN1*, herein IRE1) plays a central role in maintaining ER proteostasis [1–4]. IRE1 displays structural evolutionary conservation from yeast to primates, comprising an ER-lumenal domain, a single-pass transmembrane

**Data availability statement:** The code utilized to perform the bulk RNA sequencing analyses can be found in the following repository: https://github.com/PechuanLab/pubs-repo/tree/main/ire1xbp1 and in Zenodo (https://doi.org/10.5281/zenodo.14624658). RNA sequencing data and results have been deposited to GEO database with dataset identifier GSE285981. Proteomics data, database search results, quantification data, R script and results for statistical analysis have been deposited to ProteomeXchange Consortium via the MassIVE partner repository with the dataset identifier MSV000093902 (https://massive.ucsd.edu/ProteoSAFe/dataset.jsp?task=4c3580f-6d32644e582bb3afa2a70a83d). Original raw immunoblot images used for Figures can be found in S1 Raw Images, while raw data used to generate graphs can be found in S1 Data. The FCS files associated with the FACS experiments have been deposited in Zenodo (https://zenodo.org/records/14928071).

**Funding:** The author(s) received no specific funding for this work.

**Competing interests:** The authors have declared that no competing interests exist.

**Abbreviations:** ASO, antisense oligonucleotide; BAC, bacterial artificial chromosome; CDK, cyclin-dependent kinase; CFSE, carboxyfluorescein succinimidyl ester; CM, conditioned media; DDR, DNA damage response; Dox, doxycyclin; ELISA, enzyme-linked immunosorbent assay; EM, electron microscopy; ER, endoplasmic reticulum; EV, empty vector; FDR, false discovery rate; GO, Gene ontology; GSEA, Gene Set Enrichment Analysis; IB, immunoblot; IF, immunofluorescence; IRE1, inositol-requiring enzyme 1α; KI, kinase inhibitor; MM, multiple myeloma; NE, nuclear envelope; NTC, non-targeting control; PFA, paraformaldehyde; PI, propidium iodide; RI, RNase inhibitor; RIDD, regulated IRE1-dependent decay; SPE, solid-phase extraction; STR, short tandem repeat; ULA, ultra-low attachment; UPR, unfolded protein response.

domain, and a cytoplasmic kinase-endoribonuclease module [5]. Perturbations in ER-mediated protein folding drive IRE1 activation through homo-oligomerization, kinase-mediated *trans*-autophosphorylation, and allosteric RNase engagement. In turn, IRE1's RNase activity triggers the generation of the transcription factor XBP1s via endomotif-based intron excision followed by nonconventional mRNA splicing [1]. Furthermore, IRE1's RNase mediates the degradation of select RNAs through an endomotif-directed cleavage called regulated IRE1-dependent decay (RIDD) [6,7], as well as a more promiscuous RNA decay dubbed RIDD lacking endomotif (RIDDLE) [8]. In addition to phosphate transfer between neighboring protomers within IRE1 homo-oligomers [8–11], IRE1's kinase moiety can *trans*-phosphorylate other substrates, such as the RNA-binding proteins Pumilio [12] and FMRP [13]. Furthermore, IRE1 can physically interact with several cellular proteins, including BiP [14,15], ERDJ4 [16], PDIA6 [17], the Sec61/63 translocon [18], ribosomes [19], TRAF2 [20], non-muscle myosin [21], and Filamin A [22,23]. In higher eukaryotes, IRE1 cooperates with two other ER-resident mediators, namely PERK and ATF6 [1–4], to orchestrate the unfolded protein response (UPR). The UPR helps maintain cellular proteostasis by adjusting the ER's protein-folding capacity to its biosynthetic load, as well as via ER-associated degradation of misfolded proteins.

Cancer cells can co-opt the UPR to mitigate ER stress caused by excessive protein-folding requirements, mutations, aneuploidy, and harsh microenvironments [24–27]. Supporting this notion, cell lines derived from several malignancies, including multiple myeloma (MM) [28–33], chronic lymphocytic leukemia [34], breast cancer [35–38], and colon cancer [39–42], exhibit significant dependency on IRE1. Studies to date have attributed this requirement to IRE1's enzymatic activation of XBP1s [28–33,35,38,43,44] and of RIDD[8,45,46]. Here, we show that some cancer cells possess an unexpected nonenzymatic dependency on IRE1.

## Results

### Growth of certain cancer cell lines requires IRE1 but not its enzymatic activity

To characterize IRE1 dependency in tumors, we genetically or pharmacologically disrupted IRE1 function in a range of cancer cell lines and assessed their growth in vitro and in vivo (Fig 1A). For shRNA-based knockdown, we used doxycyclin (Dox)-induced expression of three tandem hairpins per mRNA target (see Methods). Consistent with other models [8,30], Dox-induced expression of IRE1 shRNAs (shIRE1) led to dose-dependent depletion of both IRE1 and XBP1s, while induction of XBP1 shRNAs (shXBP1) diminished XBP1s but not IRE1 (S1A Fig). Expectedly, IRE1 silencing upregulated the RIDD target CD59 [47], whereas XBP1 knockdown decreased CD59 levels below baseline, consistent with a known enhancement of RIDD in the absence of XBP1 [35,48,49]. Strikingly, IRE1 silencing strongly attenuated AMO1 cell proliferation in vitro and caused a complete regression of pre-established AMO1 tumor xenografts in vivo, whereas XBP1 silencing had little effect in both settings (Fig 1B and 1C). The differential dependency on IRE1 *versus* XBP1 suggested that AMO1 cells require IRE1 either for another enzymatic function such

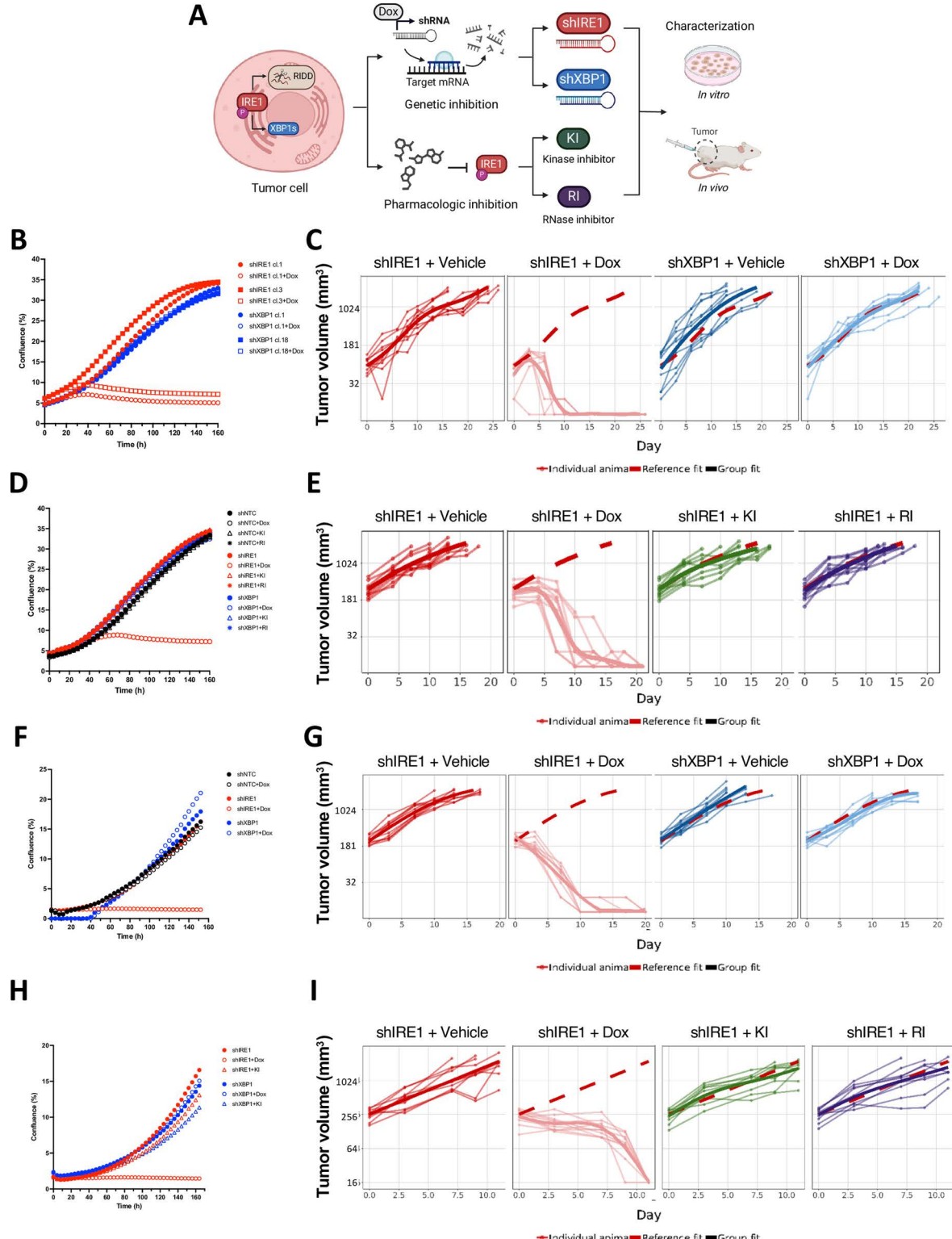

**Fig 1. Growth of certain cancer cell lines requires IRE1 but not its enzymatic activity. (A)** Schematic of the characterization of IRE1 dependency in tumor cells using the genetic and pharmacologic disruption of IRE1 or its downstream target XBP1. Created in BioRender. Zuazo-Gaztelu, I. (2025) https://BioRender.com/x33w156. **(B)** Effect of IRE1 or XBP1 knockdown on in vitro proliferation of AMO1 cells. Cells were stably transfected with

plasmids encoding doxycycline (Dox)-inducible short hairpin RNAs (shRNAs) (3 distinct shRNAs in tandem) against either IRE1 (red) or XBP1 (blue) and grown in the absence (closed symbols) or presence (open symbols) of Dox (0.2 µg/ml). Proliferation, depicted as % confluence, was monitored by time-lapse microscopy in an Incucyte instrument. Two independent clones (circles or squares) for each gene knockdown were characterized. Points are means of 15 technical replicates. Representative plot of three independent experiments shown. **(C)** Effect of IRE1 or XBP1 knockdown on in vivo growth of AMO1 tumor xenografts. C.B-17 SCID mice were implanted subcutaneously with $10 \times 10^6$ AMO1 shIRE1 cl.1 or shXBP1 cl.1 cells and allowed to form palpable tumors. Mice were then randomized into treatment groups (n = 10/group) and given either vehicle (5% sucrose) or Dox (0.5 mg/ml in 5% sucrose) in drinking water ad libitum. Tumor growth was monitored every 2–3 days until the last Vehicle-control animal reached endpoint (day 26). Thin lines represent individual mice, whereas thick lines represent group fit, corresponding to the mean growth of each group. The dashed red represents the reference fit of the vehicle group, embedded for comparison in all graphs. **(D)** Effect of IRE1 or XBP1 knockdown or enzymatic IRE1 inhibition on in vitro proliferation of AMO1 cells. Cells stably transfected with Dox-inducible shRNAs against non-targeting control (NTC, black) or IRE1 (red) or XBP1 (blue) were grown in the absence (closed symbols) or presence (open symbols) of Dox (0.2 µg/ml, circles), without or with IRE1 kinase inhibitor (KI, 3 µM, triangles) or RNase inhibitor (RI, 3 µM, asterisks). Proliferation was analyzed as in A. Data points are means of 5 technical replicates. Representative plot of three independent experiments shown. **(E)** Effect of IRE1 knockdown or enzymatic inhibition on in vivo growth of AMO1 tumor xenografts. C.B-17 SCID mice were implanted subcutaneously with $10 \times 10^6$ AMO1 shIRE1 cl.1 cells and allowed to form palpable tumors. Mice were then randomized into treatment groups (n = 10/group) and given either vehicle (5% sucrose) or Dox (0.5 mg/ml in 5% sucrose) ad libitum, or treated orally bidaily with either vehicle, or IRE1 KI (250 mg/kg) or IRE1 RI (100 mg/kg). Tumor growth was monitored until the last Vehicle-control animal reached endpoint (day 19) and plotted as in B. **(F)** Effect of IRE1 or XBP1 knockdown on in vitro proliferation of KMS27 cells. Cells stably transfected with plasmids encoding Dox-inducible shRNAs against either IRE1 (red) or XBP1 (blue) or NTC (black) were grown, treated, and analyzed as in A. One independent clone was characterized for each gene (cl.9 for shIRE1 and cl.13 for shXBP1). Data points are means of 15 technical replicates. Representative plot of three independent experiments shown. **(G)** Effect of IRE1 or XBP1 knockdown on in vivo growth of KMS27 tumor xenografts. C.B-17 SCID mice were implanted subcutaneously with $10 \times 10^6$ KMS27 shIRE1 cl.9 or shXBP1 cl.13 cells and allowed to form palpable tumors. Mice were then randomized, treated, and monitored as in B for 21 days. Tumor growth was plotted as in B. **(H)** Effect of IRE1 or XBP1 knockdown or enzymatic IRE1 inhibition on in vitro proliferation of KMS27 cells. Cells stably transfected with plasmids encoding Dox-inducible shRNAs against IRE1 (red) or XBP1 (blue) were grown in the absence (closed symbols) or presence (open symbols) of Dox (0.2 µg/ml, circles), without or with IRE1 KI (KI2, 3 µM, triangles). Proliferation was analyzed as in A. Data points are means of 6 technical replicates. Representative plot of three independent experiments shown. **(I)** Effect of IRE1 knockdown or enzymatic inhibition on in vivo growth of KMS27 tumor xenografts. C.B-17 SCID mice were implanted subcutaneously with $10 \times 10^6$ KMS27 shIRE1 cl.9 cells and allowed to form palpable tumors. Mice were then randomized, treated, and monitored as in D. Tumor growth was monitored during 12 days and plotted as in B. See also S1 Fig. All raw data can be found in S1 Data.

as RIDD—or nonenzymatically. To discern this, we blocked IRE1's catalytic activity with an IRE1-specific small-molecule kinase inhibitor (KI) [50,51] or RNase inhibitor (RI) [52] (Fig 1A). Similar to Dox-induced IRE1 silencing, treatment of AMO1 cells with either inhibitor decreased XBP1s and increased RIDD activity as measured by *DGAT2* [8,53] mRNA accumulation, similar to Dox-induced IRE1 silencing both in vitro (S1B and S1C Fig) and in vivo (S1D and S1E Fig). XBP1 knockdown in vivo also efficiently depleted XBP1s and suppressed *DGAT2* mRNA (S1F and S1G Fig).

Nevertheless, despite blocking IRE1's enzymatic activity effectively, neither IRE1 inhibitor attenuated proliferation in vitro or tumor growth in vivo of AMO1 cells—similar to the non-targeted shRNA control (shNTC), and in sharp contrast to a robust growth disruption by IRE1 knockdown (Fig 1D and 1E). Furthermore, the combination of either one of two different IRE1 KIs with the IRE1 RI also did not affect AMO1 growth (S1H and S1I Fig). To further validate the nonenzymatic dependency on IRE1, we introduced a homozygous "kinase-dead" mutation (D688N) into the endogenous IRE1 gene locus in AMO1 shIRE1 cl.1 cells via CRISPR/Cas9. D688N cells showed IRE1 protein expression but no IRE1 phosphorylation or XBP1 splicing, confirming the expected kinase-RNase disruption of IRE1 (S1I Fig). Importantly, IRE1 D688N cells proliferated similarly to IRE1 WT cells, and Dox-induced depletion of either the WT or mutant IRE1 led to a similar loss of proliferation (S1I Fig). Thus, this enzymatically inactive mutant of IRE1 enables AMO1 proliferation comparably to WT IRE1. Experiments with human KMS27 MM cells also revealed a strong nonenzymatic dependency on IRE1 (Fig 1F–1I and S1J–S1M Fig). This was also true for the other MM cell lines JJN3 and L363 and the colon carcinoma cell line HCT116 (S1N–S1S Fig). In contrast, knockdown of either IRE1 or XBP1 attenuated proliferation of OPM2 MM cells (S1T and S1U Fig), in keeping with previous evidence for the reliance of OPM2 cells on IRE1's enzymatic activity [30].

We used three additional strategies to further verify that growth disruption upon shRNA-based IRE1 silencing was not an off-target effect. First, we tested a non-overlapping siRNA, which maps to a distinct region of the IRE1 coding sequence as compared to the triple shRNAs 7, 8, and 9 used throughout the study (S1V Fig). Upon transfection into HCT116 cells, the IRE1 siRNA led to a substantial loss of cell viability as compared to a non-targeted control siRNA, in

conjunction with a marked depletion of the IRE1 protein (S1W Fig). Second, we tested an antisense oligonucleotide (ASO) which targets yet another region of the IRE1 sequence (S1V Fig). Upon transfection into HCT116 cells, the IRE1 ASO led to a substantial loss of cell viability as compared to a non-targeted control ASO, again in conjunction with a marked depletion of the IRE1 protein (S1X Fig). Third, we generated bacterial artificial chromosome (BAC) vectors containing an shRNA-resistant cDNA encoding either WT IRE1 or an RNase-dead K907A mutant of IRE1. To drive cDNA transcription without massive overexpression, we included in each plasmid 2 kb of 5′ untranslated sequence from the *IRE1* gene locus. We then performed transient transfection of each IRE1 BAC, or an "empty" vector (EV) lacking inserted IRE1 sequences, into HCT116 shIRE1 cells. We examined the ability of these constructs to rescue the loss of viability conferred by Dox-induced shRNA knockdown of endogenous IRE1. As compared to EV, both WT IRE1 and the K907A mutant substantially and significantly, albeit incompletely, rescued cell viability in the context of Dox-induced IRE1 silencing (S1Y Fig). Potential limitations of BAC-mediated transient expression include the extragenomic location of the plasmid and a possibility for mislocalization or misfolding of the encoded protein. Given these caveats, the substantial rescue we observed confirms that on-target IRE1 depletion rather than off-target shRNA activity inhibits proliferation upon shIRE1 induction. Furthermore, the ability of RNase-dead IRE1 to rescue viability as well as WT IRE1 validates the nonenzymatic nature of this IRE1 requirement. Thus, certain cancer cell lines possess a nonenzymatic dependency on IRE1.

### IRE1 depletion induces cell cycle arrest independently of apoptosis

Growth deficiency upon IRE1 depletion could stem from diminished proliferation or increased cell death. To discern this, we tracked DNA content as an indicator of cell cycle progression or apoptosis by flow cytometric analysis of AMO1 cells stained with propidium iodide (PI). Knockdown of IRE1 but not XBP1 increased the proportion of cells in the G1 phase of the cell cycle while decreasing abundance in the S and G2/M phases by 72 h (Fig 2A), suggesting a G1 arrest. Kinetic studies further revealed that most of the G1 accumulation occurred already by 24 h after Dox addition (Fig 2B). Albeit with some delay, IRE1 knockdown also increased the incidence of cells with sub-diploid (SubG1) DNA content (Fig 2A and 2B), suggesting apoptosis induction. Consistent with the proliferation data (Fig 1), neither XBP1 silencing nor enzymatic IRE1 inhibition disrupted cell cycle progression (Fig 2B and 2C). Furthermore, AMO1 cells harboring D688N mutant IRE1 showed unabated cell cycle progression similar to AMO1 cells possessing WT IRE1, while Dox-induced IRE1 depletion led to G1 accumulation in both cases (S2A Fig). To track the number of completed cell divisions versus time, we performed a flow cytometric analysis of cells stained with carboxyfluorescein succinimidyl ester (CFSE). By 48 h the vast majority of control cells had completed two divisions (generation 3) versus one (generation 2), whereas fewer IRE1-depleted cells had completed two divisions as compared to one (Fig 2D).

In keeping with the PI-based evidence for apoptosis induction, IRE1—but not XBP1-silenced cells showed dramatic cleavage of caspase-3 by 48 h, which could be blocked by the pan-caspase inhibitor Q-VD-Oph (QVD) (Fig 2E). Importantly, QVD addition during IRE1 silencing inhibited sub-G1 DNA generation yet it did not prevent the disruption of cell cycle progression, nor did it restore proliferation (Fig 2F and S2B). Earlier work implicated signaling through the JNK pathway downstream to IRE1 [20]. To examine whether the disruption of JNK signaling affected the growth of AMO1 cells, we tested the well-established JNK-specific inhibitor Tanzisertib [54]. At 1–10 μM, Tanzisertib strongly inhibited JNK phosphorylation in AMO1 cells (S2C Fig). However, in contrast to Dox-induced IRE1 silencing, Tanzisertib failed to inhibit proliferation or induce G1 accumulation, although it did have a modest anti-apoptotic effect as indicated by a diminished sub-G1 fraction in its presence (S2D and S2E Fig). Thus, it is unlikely that diminished JNK signaling downstream to IRE1 mediates the growth inhibitory effect of IRE1 silencing. As in AMO1 cells, knockdown of IRE1, but not XBP1, induced G1 accumulation and apoptosis also in KMS27, JJN3, L363, and HCT116 cells (S2F–S2I Fig). Furthermore, QVD addition blocked apoptosis without restoring proliferation and cell cycle progression upon IRE1 silencing in KMS27 (S2J–S2L Fig), JJN3 (S2M Fig) or HCT116 cells (S2N Fig). By contrast to the differential dependency of these cell lines on IRE1 versus XBP1, both the OPM2 and KMS11 cell lines, previously shown to have similar growth dependency on IRE1 or XBP1 [30],

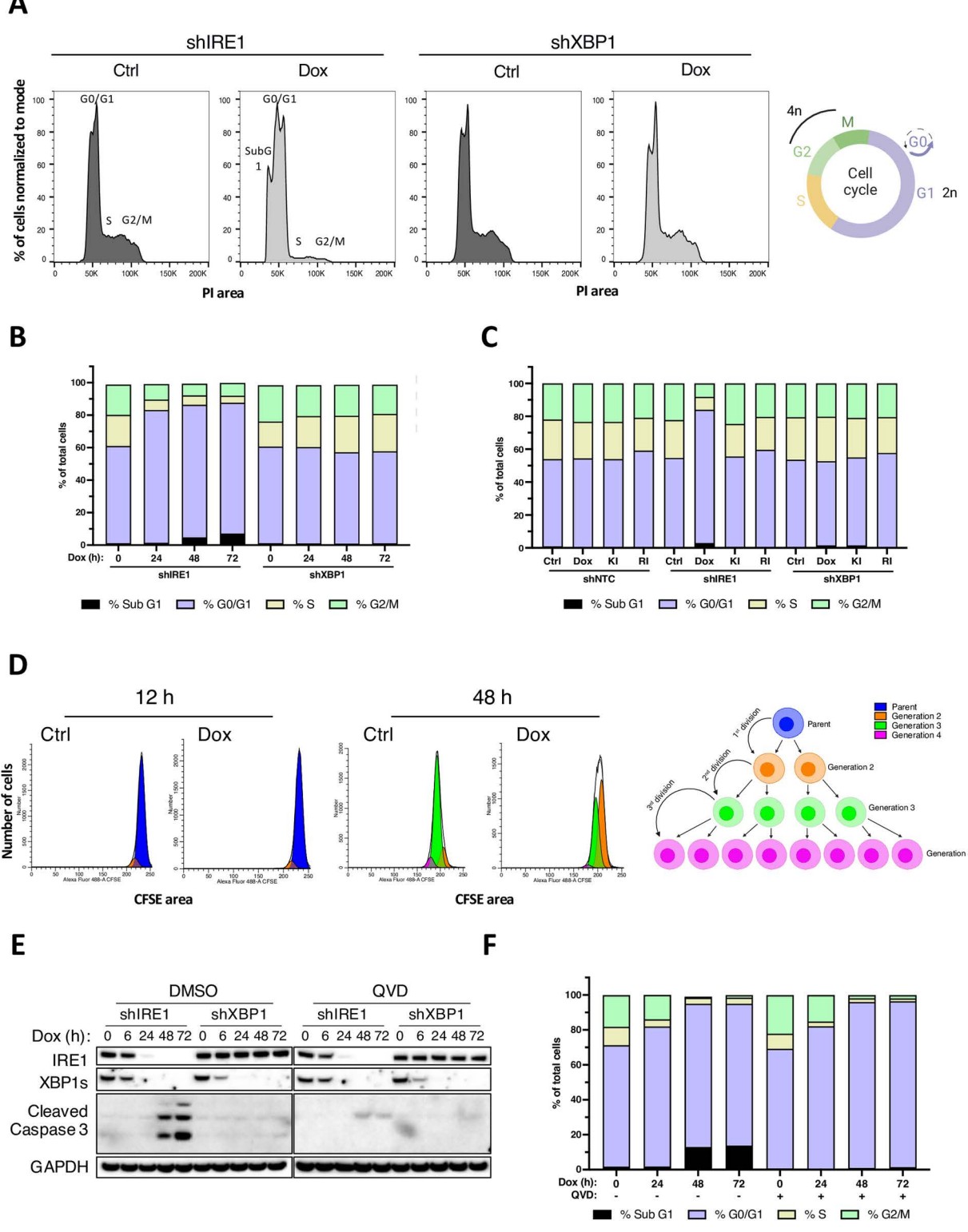

**Fig 2. IRE1 depletion induces cell cycle arrest independently of apoptosis. (A)** AMO1 shIRE1 cl.1 (left) or shXBP1 cl.1 (center) cells were incubated in the absence (dark gray) or presence (light gray) of Dox (0.2 µg/ml) for 72 h. Cells were stained with propidium iodide (PI) and analyzed by flow cytometry to determine cell cycle phases by DNA content (see schematics on the right). Created in BioRender. Zuazo-Gaztelu, I. (2025) https://

BioRender.com/u45y577. Sub-G1, G0/G1, S, and G2/M peaks are annotated for the shIRE1 histograms. Representative plot of three independent experiments shown. **(B)** AMO1 shIRE1 cl.1 or shXBP1 cl.1 cells were incubated with Dox (0.2 µg/ml) for the indicated time, analyzed as in A, and cell frequencies by cell cycle phase were depicted as stacked bar graphs. Representative plot of 3 independent experiments shown. **(C)** AMO1 shNTC, shIRE1 cl.1 or shXBP1 cl.1 were treated for 48 h with Dox (0.2 µg/ml), or with IRE1 kinase (KI) or RNase (RI) inhibitors at 3 µM. Cells were analyzed and as in A and results were graphed as in B. Representative plot of three independent experiments is shown. **(D)** Flow cytometry histograms representing AMO1 shIRE1 cell proliferation after 12 (left) or 48 (right) h culture in the presence or absence of Dox (0.2 µg/ml) measured by CFSE staining. Generations following cell division are color-coded and represented in the schematics on the right. Created in BioRender. Zuazo-Gaztelu, I. (2025) https://BioRender.com/a54m564. Representative plot of three independent experiments shown. **(E)** Validation of caspase activity and QVD-mediated inhibition. AMO1 shIRE1 cl.1 or shXBP1 cl.1 cells treated for the indicated time with Dox (0.2 µg/ml) and QVD (30 µM) and analyzed by IB. **(F)** AMO1 shIRE1 cl.1 cells were treated as in E, analyzed as in A, and graphed as in B. Representative plot of three independent experiments is shown. See also S2 Fig. All raw data can be found in S1 Data and the FCS files associated with the FACS experiments in https://zenodo.org/records/14928071.

displayed a comparable cell cycle inhibition upon knockdown of either IRE1 or XBP1 (S2O and S2P Fig). Flow cytometry of AMO1 cells stained with Annexin V and 7-AAD further confirmed the induction of apoptosis and its inhibition by QVD upon silencing of IRE1 but not under IRE1 RNase inhibition or XBP1 knockdown (S2Q and S2R Fig). To verify the uncoupling of cell cycle inhibition from apoptosis, we generated AMO1 cells with CRISPR-mediated gene disruption of caspase-8—a protease implicated in apoptosis initiation by ER stress that is functionally suppressed by IRE1 [48,55–57]. As expected, caspase-8 knockout strongly dampened caspase-3/7 and apoptosis activation upon IRE1 depletion; however, it afforded only a modest growth restoration (S2S–S2U Fig). Thus, IRE1 silencing causes cell cycle inhibition independently of apoptosis activation.

IRE1's enzymatic activity can facilitate cellular secretory function via XBP1s and RIDD [1–3]. We therefore tested whether conditioned media (CM) from IRE1-proficient cells could rescue proliferation of IRE1-depleted cells. Transfer of CM from IRE1-expressing AMO1 cells to naïve cells during subsequent IRE1 silencing led to a minor increase in proliferation during a 7-day study; however, most of the growth inhibition persisted (S2V Fig). Conversely, CM from IRE1-silenced cells did not significantly attenuate growth of IRE1-proficient cells (S2W Fig). Thus, the principal requirement for IRE1 in AMO1 cells is independent of secretory function.

### IRE1 silencing downregulates mRNA expression of numerous cell cycle genes

To follow the consequences of IRE1 silencing, we performed RNA sequencing transcriptomic analysis of AMO1 cells subjected to IRE1 *versus* XBP1 knockdown (two clones per gene in triplicates) after 0, 24, 48, and 72 h of Dox addition (S3A and S3B Fig). IRE1 or XBP1 knockdown showed expected effects on the XBP1s transcriptional targets SEC61A and SYVN1 and the RIDD mRNA targets *DGAT2* and *BCAM* [8] (S3C Fig). An exploratory KEGG pathway analysis of genes that showed significantly different expression upon IRE1 knockdown indicated two major gene categories that were most notably downregulated: *Protein Processing in the ER*, and *Cell Cycle* (Fig 3A). Further dissection revealed markedly attenuated expression of genes involved in mediating the S (Fig 3B and S3D Fig) as well as G2/M phases (Fig 3C and S3E Fig). In keeping with the flow cytometry data (Fig 2B), many of the transcriptional changes in cell cycle control genes occurred already within 24 h of Dox addition. Hallmark Gene Set Enrichment Analysis (GSEA) revealed significant downregulation of *E2F*, *Myc*, *G2/M*, and *Mitotic Spindle Gene Targets* (Fig 3D), while Gene ontology (GO) GSEA further indicated depletion of mRNAs linked to *DNA Replication*, *Chromosomal Region*, *Chromosome Centromeric Region*, and *Chromosome Segregation* (Fig 3E). Thus, silencing of IRE1, but not of XBP1, decreased transcription of many cell cycle genes involved in chromosome replication, organization, and segregation.

### IRE1 depletion engages the TP53 pathway and specific CDK inhibitors

Amongst gene sets that were significantly impacted by IRE1 silencing was the TP53 (p53) tumor suppressor pathway, which showed elevated expression of most of its target-gene clusters (S3F and S3G Fig). To complement our

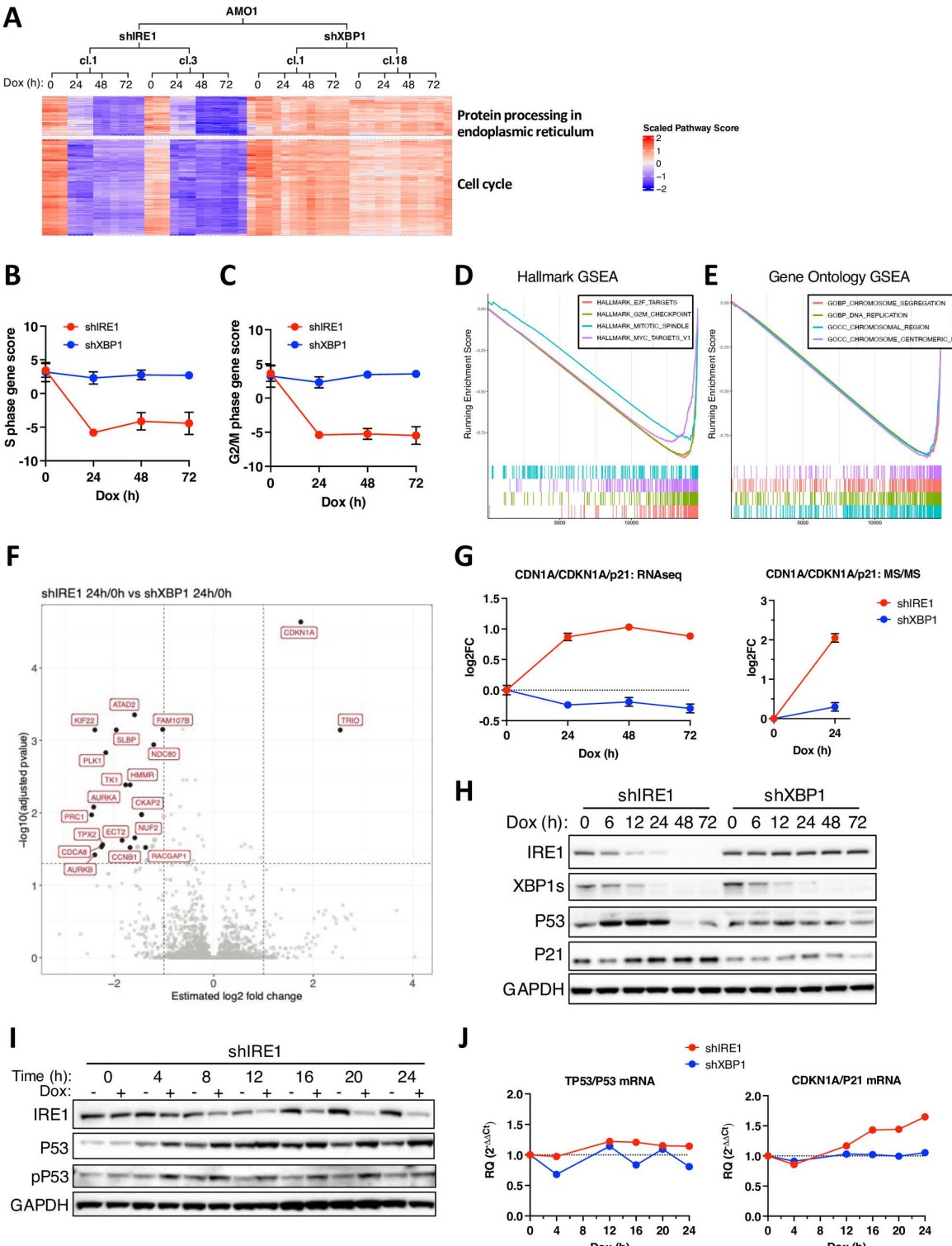

**Fig 3. IRE1 silencing downregulates cell cycle genes and engages TP53 and specific CDK inhibitors. (A)** Effect of IRE1 or XBP1 knockdown on mRNA expression of genes involved in Protein Processing in the ER or in cell cycle control. AMO1 shIRE1 cl.1 or cl.3 cells and shXBP1 cl.1 and cl.18 cells were incubated for the indicated time with Dox (0.2 μg/ml) in triplicates and subjected to bulk RNA sequencing. Shown are heat maps indicating scaled expression of distinctly regulated mRNAs encoded by KEGG pathway gene sets. **(B)** Effect of IRE1 or XBP1 knockdown on mRNA expression of

genes involved in the S phase of the cell cycle, calculated as the S phase gene score. Score for samples in A shown. (C) Effect of IRE1 or XBP1 knockdown on mRNA expression of genes involved in the G2 and M phases of the cell cycle, calculated as the G2/M phase gene score. Score for samples in A shown. (D) RNAseq data for mRNAs specifically regulated upon IRE1 knockdown but not upon XBP1s at 24 h were queried by Gene Set Enrichment Analysis (GSEA). Enrichment plots from the most significantly downregulated Hallmark gene sets shown. (E) Enrichment plots as in D for Gene Ontology (GO) gene sets. (F) Effect of IRE1 vs. XBP1 knockdown on the proteome of AMO1 cells. AMO1 shIRE1 cl.1 or cl.3 cells and shXBP1 cl.1 and cl.18 cells were incubated for 24 h with Dox (0.2 µg/ml) in triplicates and analyzed for protein abundance by MS/MS. Depicted is a volcano plot showing estimated log2 fold change in protein abundance as a function of −log10 of the adjusted $p$ value, comparing 24 h over 0 h for shIRE1 vs. shXBP1. Most significantly altered proteins implicated in cell cycle regulation are labeled. (G) Effect of IRE1 or XBP1 knockdown on CDKN1A/p21 mRNA (left) and protein (right) levels. Data depicted are from RNAseq and proteomics analyses described in A and F, respectively. Data points are mean ± SE for all biological and technical replicates normalized to t = 0 h for each cell line. (H) Effect of IRE1 or XBP1 knockdown on p53 and p21 protein levels. AMO1 shIRE1 cl.1 or shXBP1 cl.1 cells were incubated for the indicated time with Dox (0.2 µg/ml) and analyzed by IB. Representative blot of 3 independent experiments shown. (I) Effect of IRE1 knockdown on p53 and phospho-p53 (pP53) levels. AMO1 shIRE1 cl.1 cells were incubated for the indicated time in the absence or presence of Dox (0.2 µg/ml) and analyzed by IB. Representative blot of three independent experiments shown. (J) Effect of IRE1 or XBP1 knockdown on p53 and p21 mRNA levels. Cells as in I were analyzed by RT-qPCR. Data points are one biological replicate normalized to its untreated counterpart. Representative plot of three independent experiments shown. See also S3 Fig. All raw data can be found in S1 Data.

transcriptomic data, we performed a comparative proteomic analysis of AMO1 cells during knockdown of IRE1 *versus* XBP1 (2 clones each). We collected cells at 0 and 24 h after Dox addition and characterized proteolytic digests of cell lysates by liquid chromatography-mass spectrometry (LC-MS/MS). While multiple factors associated with cell cycle progression were markedly downregulated (see below), the most significantly upregulated protein was the cyclin-dependent kinase (CDK) inhibitor CDKN1A (p21) ($p = 2.28 \times 10^{-5}$), which also showed a clear increase at the mRNA level ($p = 2.31 \times 10^{-16}$) (Fig 3F and 3G). Immunoblot (IB) analysis indicated accumulation of p21 by 12 h after Dox addition, while upregulation of the p53 protein occurred earlier, beginning at 6 h (Fig 3H). While p53 levels later declined, p21 levels remained stable. QVD treatment prolonged the upregulation of p53 through 48 h (S3H Fig), suggesting that the decline in p53 may be due to caspase-mediated cleavage [58]. The early gain in p53 was accompanied by phosphorylation on Ser15 (Fig 3I), suggesting functional p53 engagement [59]. RT-qPCR analysis further showed that the upregulation of p53 was non-transcriptional while p21 induction was evident at the mRNA level (Fig 3J), in keeping with the known transcriptional p21 activation by p53 [60]. The proteomic data also suggested that IRE1 silencing upregulated the CDK inhibitor CDKN1B/p27 (S3I Fig), albeit without statistical significance ($p = 0.116$). Further analysis by IB provided additional evidence that IRE1 knockdown upregulated both p21 and p27 in AMO1 and KMS27 cells (S3J Fig). In contrast to p21, p27 did not show enrichment at the mRNA level (S3K Fig), supporting a non-transcriptional regulation. Given that DNA damage often drives p53 activation, we interrogated specific markers of the DNA damage response (DDR) known to operate upstream to p53 upon detection of DNA double-strand breaks [61]. Knockdown of IRE1 but not XBP1 induced time-dependent upregulation and phosphorylation of histone H2AX (S3L Fig); it also increased 53BP1 levels in concert with the engagement of p53 (S3M Fig), suggesting that IRE1 depletion augments DNA damage. Thus, IRE1 knockdown induces the TP53 transcriptional program and leads to p21 accumulation.

## IRE1 silencing increases chromosome instability

The proteomic analysis also revealed significant changes in the levels of a number of factors annotated as regulating cell cycle or proliferation. These included proteins linked to G1/S, i.e., ATAD2, TK1, and SLBP; mitosis, i.e., KIF22, PLK1, NDC80, HHMR, AURKA, CKAP2, INCENP, TPX2, CDC8, NUF2, AURKB, and CCNB1; or cytoskeletal coordination during the cell cycle, i.e., PRC1, ECT2, and RACGAP1, and TRIO (Fig 3F). Hallmark GSEA underscored *E2F Targets*, *G2/M Checkpoint*, and *UV Response to DNA Damage* (S4A Fig); and GO GESA highlighted *Cell division*, *Chromosome Organization*, and *Chromosome Segregation* (S4B Fig). Whereas more than half of the proteins modulated by IRE1 knockdown are annotated as nuclear, most proteins affected by XBP1 silencing are non-nuclear (S4C Fig).

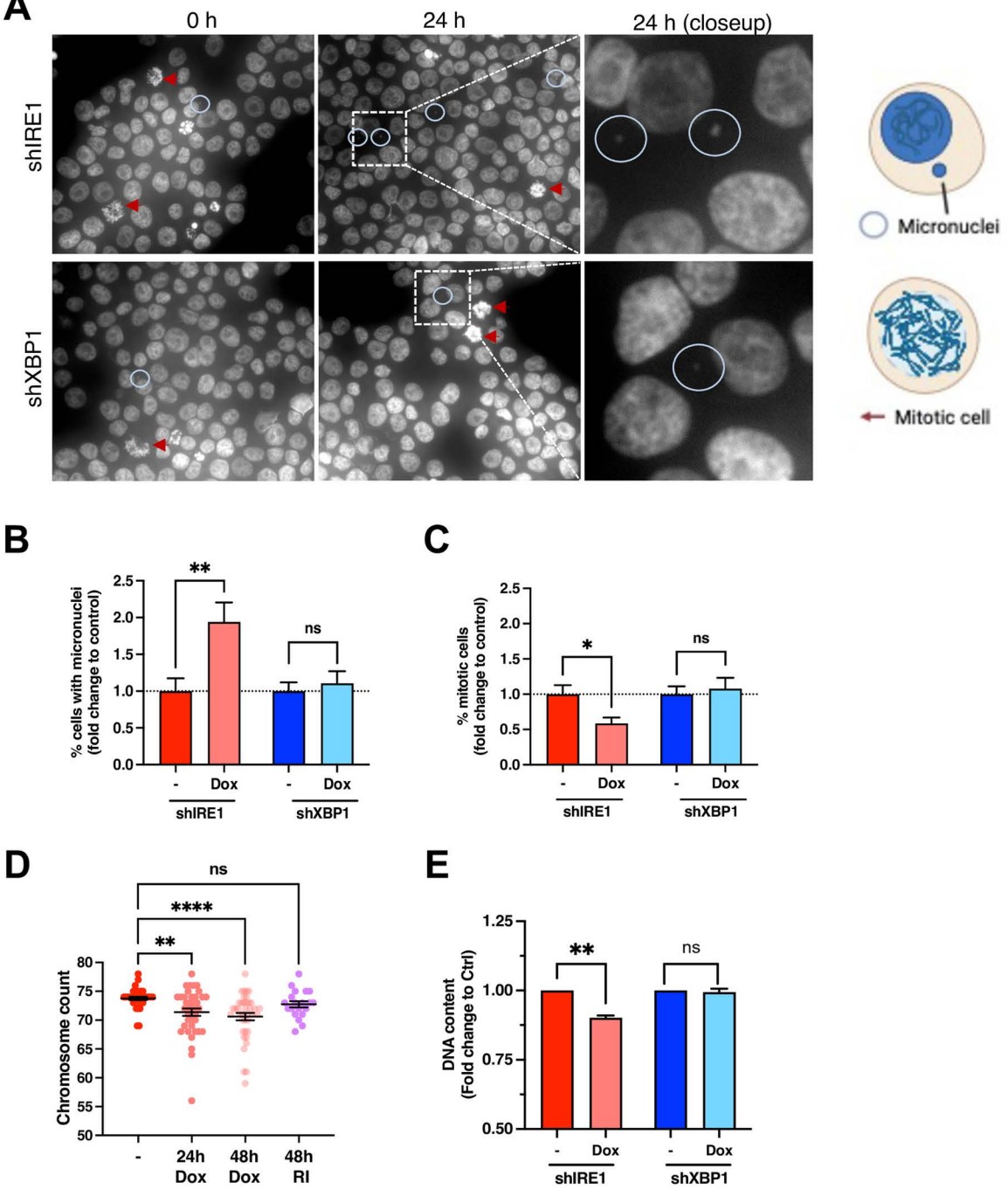

**Fig 4. IRE1 depletion increases chromosome instability. (A)** Effect of IRE1 or XBP1 knockdown on frequency of micronuclei and mitotic events. AMO1 shIRE1 cl.1 and shXBP1 cl.1 were cultured in the presence or absence of Dox (0.2 μg/ml) for 24 h, stained with Hoechst DNA dye, and analyzed by fluorescence microscopy. Images representative of at least 10 fields examined per condition of three independent experiments. Blue circles indicate examples of cells with micronuclei, while red arrows indicate mitotic cells. A schematic of both is shown on the right. Created in BioRender. Zuazo-Gaztelu, I. (2025) https://BioRender.com/a54w687 Scale bar = 20 μm. **(B)** Quantification of annotated events in A normalized per total amount of cells per field. Ten fields per condition with at least 100 cells were counted for $n = 3$ independent experiments. Mean ± *SEM* of the fold change to untreated controls for each cell line. Mann–Whitney *U* statistical test. **(C)** Quantification of mitotic cells exemplified in A per total amount of cells per field. Ten fields per condition with at least 100 cells were counted for three independent experiments. Mean ± SEM of the fold change to untreated controls for each cell line. Mann–Whitney *U* statistical test. **(D)** Effect of IRE1 knockdown on chromosome numbers. AMO1 shIRE1 cl.1 cells were incubated for the indicated time with Dox (0.2 μg/ml) or with RI (3 μM) and metaphase spreads were prepared and quantified for chromosome count. Each datapoint indicates one

metaphase spread. Mean ± SEM for three independent experiments. Kruskal-Wallis with Dunn's multiple comparisons statistical test. **(E)** Effect of IRE1 or XBP1 knockdown on DNA content. AMO1 shIRE1 cl.1 and shXBP1 cl.1 were treated for 24 h with QVD (30 µM) in the presence or absence of Dox (0.2 µg/ml) or RI (3 µM), stained with the DNA dye Hoechst, and analyzed by flow cytometry. Fold change to untreated controls of the DNA content per total amount of live cells is depicted. Mean ± SEM for five independent experiments for shIRE1 and three for shXBP1. Mann–Whitney $U$ statistical test. *$p < 0.05$; **$p < 0.005$; ***$p < 0.001$; ****$p < 0.00001$ consensus. A $p$ value > 0.05 was considered non-significant (ns). See also S4 Fig. All raw data can be found in S1 Data.

Given the apparent impact of IRE1 depletion on factors regulating chromosome dynamics during the cell cycle, we investigated chromosome integrity by assessing Hoechst-stained AMO1 cells for the presence of micronuclei by fluorescence microscopy. IRE1 silencing significantly increased the relative frequency of micronuclei, in contrast to XBP1 knockdown (Fig 4A and 4B; S4D Fig), suggesting an exacerbated chromosomal instability [62]. In concert, knockdown of IRE1 but not of XBP1 also decreased the relative frequency of mitotic cells (Fig 4C). We obtained similar evidence of an increased abundance of micronuclei and a decreased mitotic cell count upon IRE1 knockdown in KMS27 cells (S4E and S4F Fig). To further assess chromosomal stability, we performed a cytogenetic analysis of G-banded metaphase spreads of AMO1 cells. While we did not discern consistent changes in chromosome translocations, we observed a significant drop in chromosomal count in IRE1-silenced cells. Specifically, AMO1 cells had a mean chromosome count of 73.8 ± 0.3 at baseline (Fig 4D), consistent with their known aneuploidy [63]. Strikingly, IRE1-depleted cells lost approximately 2 chromosomes on average by 24 h and one additional chromosome by 48 h, whereas cells treated in parallel with IRE1 RI for 48 h did not display a significant chromosome loss. Silencing of IRE1 but not of XBP1 also led to a significant decrease in DNA content (Fig 4E), in keeping with the evident increase in micronuclei and drop in chromosome count. Together, these results suggest a nonenzymatic involvement of IRE1 in supporting the integrity and correct trafficking of chromosomes during cell division.

### IRE1 depletion decreases heterochromatin, DNA and H3K9me3 methylation, and UHRF1

To further interrogate chromatin changes, we examined nuclear morphological features of AMO1 cells by electron microscopy (EM). Based on electron density and distribution features, we could readily discern heterochromatin, evident from clusters of electron-dense material; nucleoli; and euchromatin, indicated by dispersed patches of relatively electron-lucent material (Fig 5A). At 24 h after Dox addition, IRE1-silenced cells showed a marked reduction in heterochromatin, whereas XBP1-depleted cells showed similar heterochromatin density to that of baseline controls (Fig 5A and S5A Fig). Quantification of multiple fields indicated that IRE1 knockdown significantly decreased the proportion of cells with heterochromatin and increased that of cells with euchromatin, whereas XBP1 silencing did not significantly alter these proportions as compared to controls (Fig 5B). A blind independent EM study at a different laboratory (University of Utrecht), also indicated a decrease in cellular heterochromatin and an increase in euchromatin in response to IRE1 silencing (S5B Fig). Thus, IRE1 depletion attenuates DNA and H3 methylation and heterochromatin abundance.

To examine additional aspects of chromatin modulation, we measured global DNA methylation using a 5-methylcytosine enzyme-linked immunosorbent assay (ELISA). IRE1 knockdown significantly decreased DNA methylation levels, whereas XBP1 depletion did not (Fig 5C). We also interrogated histone H3 methylation by IB against H3 and H3K9me3. Although we did not detect H3Kme9 at baseline, treatment with the DNA methylation inhibitor 5-azacytidine (5-aza) enriched this H3 trimethylation mark, while IRE1 knockdown attenuated this increase (Fig 5D and S5C Fig). Whereas 5-aza alone had a weak anti-proliferative effect on AMO1 cells, its combination with silencing of IRE1 but not of XBP1 led to a more complete growth inhibition than did IRE1 disruption alone (Fig 5E and S5D Fig). In light of the observed heterochromatin reduction in response to IRE1 silencing, we further analyzed the RNA-seq data with a lens toward gene sets involved in regulating this process. The GO terms *Heterochromatin Formation*, *Regulation of DNA Methylation-Dependent Heterochromatin Formation*, and *H3K9me3 Modified Histone Binding*, all showed declines in multiple genes in response to IRE1 knockdown

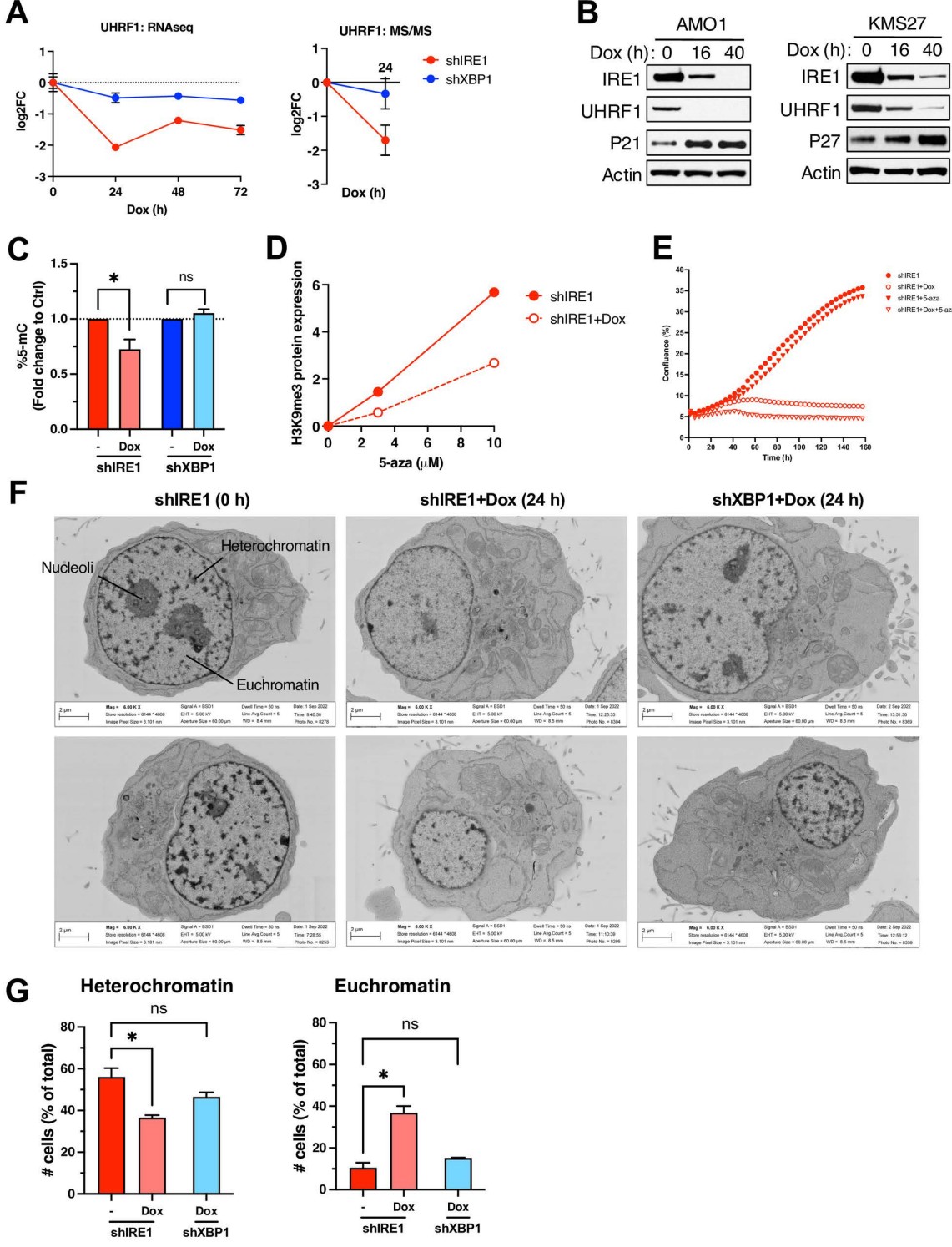

**Figure 5**

**Fig 5. IRE1 depletion decreases heterochromatin, DNA and H3K9me3 methylation, and UHRF1. (A)** Effect of IRE1 or XBP1 knockdown on heterochromatin density. AMO1 shIRE1 cl.1 and shXBP1 cl.1 cells were incubated for 24 h with Dox (0.2 μg/ml). Cells were then analyzed by electron microscopy (EM). Images depict 6000X magnification. Representative heterochromatin, euchromatin, and nucleoli are indicated in top-left image. **(B)**

AMO1 cells as in A were analyzed by EM to determine the frequency of nuclei with electron-dense heterochromatin (HC, left) or dispersed euchromatin (EC, right) per total nucleated cell count. For each sample, four fields were analyzed at 1000× magnification with approximately 30 cells per field. Mean ± SE*M*. Mann–Whitney *U* statistical test. **(C)** Effect of IRE1 knockdown on DNA methylation. AMO1 shIRE1 cl.1 and shXBP1 cl.1 were cultured in the presence or absence of Dox (0.2 µg/ml) for 24 h and gDNA was extracted. Global DNA methylation levels were quantified by 5-methylcytosine (5-mc) ELISA, normalized by gDNA amount, and shown as fold change to the untreated control. Mean ± SEM for four independent experiments for shIRE1 and 3 for shXBP1. Mann–Whitney *U* statistical test. *$p < 0.05$. A $p$ value $> 0.05$ was considered non-significant (ns). **(D)** Effect of IRE1 knockdown on H3K9 trimethylation. AMO1 shIRE1 cl.1 cells were incubated for 24 h in the absence (closed symbols) or presence (open symbols) of Dox (0.2 µg/ml) with the indicated concentration of 5-azacytidine (5-aza). H3K9me3 levels were quantified by IB against H3K9me3 as compared to total H3 levels (IB images are shown in S6C Fig; the 5-aza concentrations of 0, 3, and 10 µM are depicted in the graph). **(E)** Effect of IRE1 knockdown and 5-aza treatment on proliferation. AMO1 shIRE1 cl.1 cells were incubated for the indicated time in the absence (closed symbols) or presence (open symbols) of Dox (0.2 µg/ml), without (circles) or with (triangles) 5-aza (1 µM). Data points represent the means of five technical replicates. Representative plot of three independent experiments shown. **(F)** Effect of IRE1 or XBP1 knockdown on UHRF1 mRNA (left) and protein (right) levels. Data depicted are from RNAseq and proteomics analyses described in Fig 3A and 3F, respectively. Mean ± SE for all biological and technical replicates normalized to $t = 0$ h for each cell line. **(B)** Effect of IRE1 knockdown on UHRF1 protein levels. AMO1 shIRE1 cl.1 and KMS27 shIRE1 cl.9 cells were incubated for the indicated time with Dox (0.2 µg/ml) and analyzed by IB. Representative blots of three independent experiments. See also S5 Fig. All raw data can be found in S1 Data.

(S5E Fig). The top downregulated gene in the first and third signatures was UHRF1, a multidomain protein that promotes DNA methylation and chromatin modification and is implicated in cell cycle control in several cancers [64–66]. Consistent with the RNA-seq results, where it was significantly downregulated upon IRE1 knockdown ($p = 3.21 \times 10^{-13}$), UHRF1 also displayed a considerable decrease by MS/MS, albeit non-significant ($p = 0.61$) (Fig 5F). IB analysis of AMO1 and KMS27 cells indicated a marked downregulation of the UHRF1 protein in response to IRE1 knockdown (Fig 5G). Furthermore, tumor samples from AMO1 and KMS27 xenografts exhibited marked downregulation of UHRF1 for both cell lines, in concert with upregulation of p53, p21 (only for AMO1), and p27 (for both AMO1 and KMS27) in the context of Dox-induced IRE1 silencing but not enzymatic inhibition (S5F and S5G Fig). Thus, UHRF1 may represent a useful biomarker for tracking IRE1-dependent heterochromatin modulation.

## IRE1 silencing in synchronized cells induces cell cycle phase-specific molecular changes

IB analysis of multiple cell cycle proteins confirmed that their alteration by IRE1 knockdown was nonenzymatic, as indicated by selective changes upon depletion of IRE1, but not in the context of IRE1 enzymatic inhibition, D688N mutation, or XBP1 knockdown (S6A and S6B Fig). To discern cell cycle phase-specific proteomic changes, we synchronized AMO1 cells in the G1 phase by treatment with an established CDK4/6 inhibitor [67]. We then washed out the inhibitor to allow cycling to resume more synchronously. At $t = 0$ after CDK4/6 inhibitor washout, over 80% of control cells appeared in G1 (Fig 6A and S6C Fig)—as compared to the 60% we detected without synchronization (Fig 2B). Cells advanced into S phase by 8 h, entering G2 and M by 12–20 h (Fig 6A and S6C Fig). IRE1 knockdown during synchronization increased the proportion of cells in G1 to approximately 90% at $t = 0$, and markedly impaired transition from G1 into S and further progression into G2 and M. Relative to controls, IRE1-silenced cells showed an upregulation of γH2AX and p21 at $t = 0$, possibly due to the prior depletion of IRE1 during synchronization, and an attenuated accumulation of UHRF1 over time (Fig 6B). IRE1 knockdown also altered the kinetic profiles of several proteins involved in G1/S, including CCNA2, Rb, pRb, and TK1; as well as proteins involved in G2/M, namely CCNB1, CDC20, CDCA5, CDCA8, AURKB, Geminin, Securin, KIF22, and pH3 (Fig 6B).

Next, we synchronized AMO1 cells in late G2 by treatment with an established CDK1 inhibitor [68], followed by inhibitor washout to allow mitotic progression. Flow cytometry indicated approximately 35% of the cells in G2 and another 5% in M phase at $t = 0$ (Fig 6C and S6D Fig), totaling 40% as compared to the 20% of cells found in G2/M in non-synchronized cultures (Fig 2B). Of note, synchronization in G2/M was associated with some increase in the overall levels of apoptosis. Notwithstanding, by 90 min post washout the proportion of cells in G1 began to increase, indicating completion of mitosis and re-entry into G1. IB analysis during the first 90 min after CDK1 inhibitor washout indicated a lower level of CDC20,

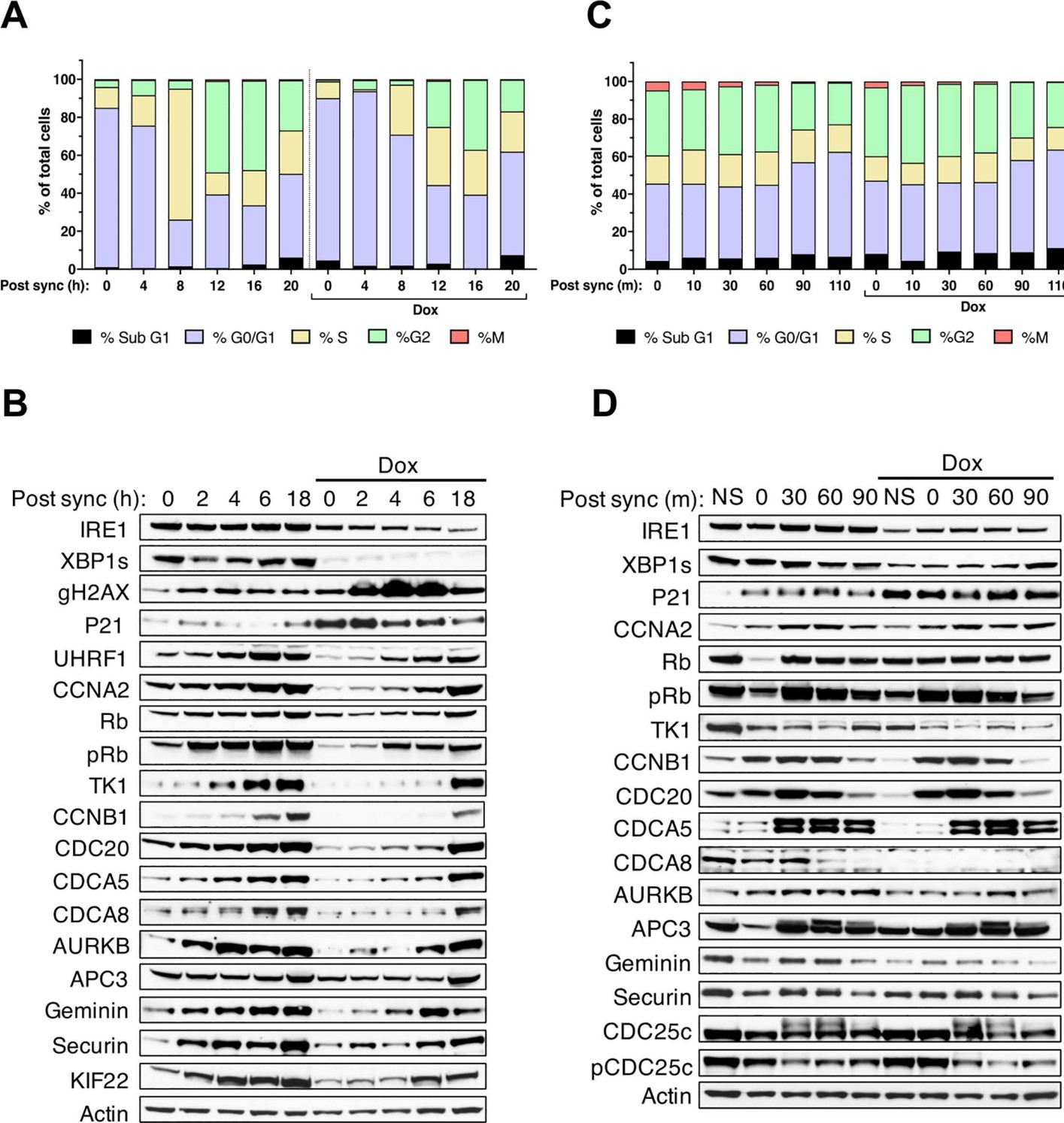

**Fig 6. IRE1 depletion downregulates multiple cell cycle proteins in synchronized cells. (A)** Effect of IRE1 knockdown during G1 synchronization on cell cycle progression. Stacked bar graphs showing cell frequencies by cell cycle phase for AMO1 shIRE1 cl.1 cells incubated with CDK4/6 inhibitor (1 μM) for 24 h in the absence or presence of Dox (0.2 μg/ml). Cells were washed to resume cycling in the absence or presence of Dox and at the indicated time points stained with PI and analyzed by flow cytometry to determine cell cycle phases by DNA content. Mitosis (M) was distinguished from G2 by phospho-H3 staining. Representative plot of 3 independent experiments shown. **(B)** Effect of IRE1 knockdown during G1 synchronization on the

abundance of cell cycle proteins. AMO1 shIRE1 cl.1 cells were synchronized as in A in the absence or presence of Dox (0.2 µg/ml), washed to resume cycling in the absence or presence of Dox, and analyzed at the indicated time points by IB. Representative blot of 3 independent experiments shown. **(C)** Effect of IRE1 knockdown during late G2 synchronization on cell cycle progression. AMO1 shIRE1 cl.1 cells were incubated with CDK1 inhibitor (9 µM) for 20 h in the absence or presence of Dox (0.2 µg/ml). Cells were washed to resume cycling in the absence or presence of Dox and at the indicated time points stained and anlyzed as in A. Representative plot of 3 independent experiments shown. **(D)** Effect of IRE1 knockdown during late G2 synchronization on abundance of cell cycle proteins. AMO1 shIRE1 cl.1 cells were synchronized as in C in the absence or presence of Dox (0.2 µg/ml), washed to resume cycling in the absence or presence of Dox and analyzed at the indicated time points by IB. Representative blot of three independent experiments shown. See also S6 Fig. All raw data can be found in S1 Data.

CDCA8, AURKB, Geminin, and pCDC25c in IRE1-depleted cells versus controls (Fig 6D). Taken together, these results suggest that IRE1 silencing affects both early and late events during the cell cycle.

### Endogenous IRE1 protein can localize to the nuclear envelope

The outer membrane of the nuclear envelope (NE) is contiguous with ER membranes [69]. We therefore reasoned that the regulation of cell cycle events might be facilitated by localization of the IRE1 protein to the NE. While one would expect IRE1 to be capable of localizing to the NE, certain mechanisms might restrict such distribution. To investigate IRE1 localization empirically, we first performed subcellular fractionation of AMO1 cell lysates followed by IB, which revealed detectable IRE1 protein in the nuclear compartment (Fig 7A). This fraction also contained the nuclear protein Lamin B1 but lacked the Golgi marker 58K, verifying its relative purity. Dox treatment depleted the IRE1 protein in the nuclear fraction to a level below detection (S7A Fig), further confirming specificity. Moreover, in contrast to Lamin B1, proteins containing the ER-localized epitope KDEL and the ER-resident protein REEP5 were not detectable in the nuclear fraction, further excluding the possibility of ER contamination. Next, to investigate IRE1's localization more directly, we leveraged a previously validated U2OS cell line expressing an endogenous IRE1 allele marked with a C-terminal HaloTag through CRISPR/CAS9-mediated gene editing [9]. Immunofluorescence (IF) microscopy of cells stained with anti-HaloTag antibody suggested that while most of the tagged IRE1 was associated with the ER, some IRE1 protein could be detected in proximity to Lamin B1, which marks the nuclear-matrix on the inner side of the NE (Fig 7B and S7B Fig). To obtain a more precise microscopic localization, we developed a sensitive immuno-EM technique to detect Halo-tagged endogenous IRE1, using AMO1 cells harboring the Dox-inducible IRE1 shRNA. IB analysis of the modified AMO1 cells verified complete HaloTag labeling of the endogenous IRE1 protein, and its depletion upon Dox addition (S7C Fig). Extending the IF results in U2OS cells, Halo-tagged IRE1 colocalized with LaminB1 in AMO1 and in HCT116 cells (S7D and S7G Fig). Immuno-EM staining with the anti-HaloTag antibody and protein A Gold[10] produced a relatively sparse signal; nevertheless, gold particles could be detected not only in association with ER membranes but also with the NE (Fig 7C and S7H Fig). Thus, some of the cellular IRE1 protein can reside in the NE, most likely in the outer membrane of this compartment.

### Discussion

Our results carry important implications for the current understanding of IRE1's biological roles, as well as for the development of more effective therapeutic strategies to disrupt IRE1.

Evidence to date has suggested that cancer-cell dependency on IRE1 principally involves IRE1's enzymatic role in the activation of XBP1s, with some possible additional contribution of RIDD [24,26,27]. Our present results demonstrate that certain cancer cells can display a surprising nonenzymatic dependency on IRE1, without a significant requirement for XBP1s or RIDD. For some tumor models, namely AMO1 and KMS27, IRE1 dependency appeared particularly strong, as IRE1 silencing caused a strong growth inhibition in vitro and a rapid and complete tumor regression in vivo. Although most of the cell lines we studied represent MM, we observed a similar nonenzymatic IRE1 requirement in HCT116 colorectal carcinoma cells, suggesting that this phenotype is not necessarily restricted to one type of cancer. Furthermore, cell lines that express either high or low levels of IRE1 mRNA displayed nonenzymatic dependency (S7I Fig), indicating that IRE1

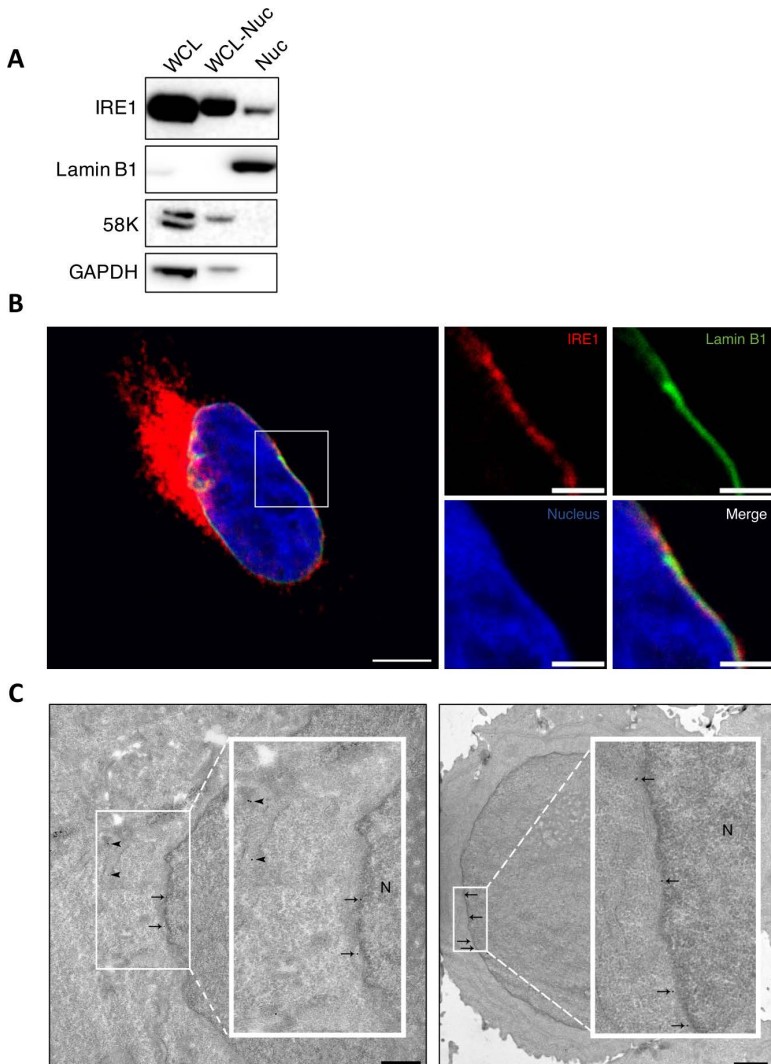

**Fig 7. Endogenous IRE1 protein can localize to the nuclear envelope. (A)** AMO1 cells were subjected to subcellular fractionation and analyzed by IB. WCL, whole-cell lysate. WCL-Nuc: Whole-cell lysate lacking the nuclear fraction. Nuc, nuclear fraction. The nuclear protein Lamin B1 and the Golgi marker 58K were used to confirm the purity of the nuclear fraction. **(B)** Detection of endogenous Halo-tagged IRE1 and non-tagged Lamin B1 by confocal microscopy. U2OS HaloTag cells, containing a C-terminal HaloTag into the endogenous IRE1 gene locus [9], were examined. Cells were cultured in the presence of Janelia 646 HaloTag ligand (green), fixed, stained with anti-Lamin-B1 (red) antibodies, and analyzed by confocal microscopy. Left: Merge field of IRE1 (red), Lamin B1 (green), and nucleus (blue) is shown. Scale bar = 5 µm. Right: Close up of the nuclear envelope where each channel is shown separately and merged. Scale bar = 2 µm. Pixel size = 48 nm. Representative image of two independent experiments shown. **(C)** Immuno-EM analysis of Halo-tagged endogenous IRE1 in AMO1 cells. A C-terminal HaloTag was inserted into the endogenous IRE1 gene locus in AMO1 cells as previously described [9]. Cells were fixed, stained with anti-HaloTag and protein-A-Gold 10 nm conjugate (protein-A-Gold10), and analyzed by EM. Arrowheads and arrows indicate gold particles detected in association with ER membranes or nuclear envelope membranes, respectively. N = nucleus. Images from two different cells are shown. See also S7 Fig.

abundance does not predict this phenotype. Aneuploidy and chromosomal instability are frequent hallmarks of malignancy [70]. Accordingly, it is conceivable that certain genome alterations such as chromosomal translocations give rise to a unique dependency on IRE1 and a specific vulnerability to its disruption. The nature of such genomic changes remains to be investigated.

Regardless, our data suggest that an unidentified scaffolding role of IRE1—mediated by one or more of its known protein–protein associations, or perhaps through some unknown interaction—supports cell cycle progression. We found that JNK inhibition did not affect growth of the IRE1-dependent cell line AMO1, excluding the JNK pathway as a central mediator of cell cycle control by IRE1. The ER plays important roles in facilitating specific events of the cell cycle, for example, the breakdown and reassembly of the nuclear envelope during mitosis [69]. Our immune-EM methodology to detect the *endogenous* IRE1 protein confirmed that some IRE1 molecules can be associated with the NE. This should enable future studies to interrogate whether IRE1 fulfills specific structural scaffolding functions at this location to facilitate ER-to-nucleus coordination during cell cycle progression. Intriguingly, previous IP-MS studies suggested that IRE1 may directly interact with proteins involved in cell cycle regulation [23].

IRE1 silencing led to a cell cycle arrest predominantly in G1 as well as to apoptotic caspase activation, the former being independent of the latter. IRE1 depletion augmented DNA damage and chromosome instability, evident by an increased abundance of specific DDR markers and of micronuclei, and by loss of chromosomes and DNA. These events may underlie the engagement of the p53 pathway, known to inhibit cell cycle progression across the G1-S as well as the G2-M checkpoints. CDKN1A/p21, found to be upregulated both at the mRNA and protein levels on IRE1 silencing, is a major transcriptional target of p53 that mediates cell cycle inhibition [59,60]. Furthermore, the non-transcriptional upregulation of CDKN1B/p27 may provide additional cell cycle inhibition upon IRE1 silencing. The p53 pathway also contributes to the induction of apoptosis in response to DNA damage [71]. Indeed, proapoptotic transcriptional targets of p53 such as *BAX* and *DR5/TNFRSF10B* [72,73] were upregulated upon knockdown of IRE1 but not XBP1 (S3G Fig), and the activation of caspase-3/7 required caspase-8 (S2O Fig)—a critical caspase downstream to DR5 [74]. A distinct interaction between the DDR and IRE1 has previously been reported based on evidence that DNA damage augments IRE1 activation and RIDD [75]. Moreover, BRCA1—an E3 ubiquitin ligase involved in DNA repair—promotes proteostasis by ubiquitinating IRE1 [76]. Together, these findings suggest reciprocal regulation between IRE1 and the DDR, wherein DNA damage enzymatically activates IRE1 while IRE1 nonenzymatically supports DNA integrity.

Another notable outcome of IRE1 knockdown was heterochromatin depletion, in association with decreased DNA methylation and 5-aza-induced H3K9me3 modification. IRE1 silencing downregulated UHRF1, which acts as an epigenetic regulator by bridging DNA methylation and chromatin modification [77]. UHRF1 specifically recognizes and binds hemi-methylated DNA at replication forks and recruits the methyltransferase DNMT1 to ensure faithful propagation of DNA methylation patterns during S phase. UHRF1 also plays a key role in chromatin modification, binding to H3K9me3 and recruiting several chromatin-associated proteins [77,78]. Whether and how the heterochromatin phenotypes are connected to the downregulation of UHRF1 remains to be investigated.

Our observations suggest that IRE1 fulfills a nonenzymatic structural function that supports cell cycle progression and is uniquely critical in certain cancer cells. More specifically, our transcriptomic and proteomic analyses indicate that IRE1 supports DNA replication, chromosome congression and segregation, as well as chromatin maintenance. IRE1 silencing downregulated factors involved in advancing the G1/S as well as G2/M phases. It is possible that the inhibition of S-phase causes mitotic disruptions within the same cell cycle. Likewise, perturbations in chromosome organization or movement during mitosis may lead to a G1 arrest during the subsequent cycle of afflicted daughter cells [79].

Perhaps related to the present findings, evidence from budding yeast suggests that IRE1 supports the fidelity of mitotic chromosome segregation in *S. cerevisiae* [80]. Further studies reveal the existence of an ER stress surveillance mechanism that ensures proper ER inheritance during yeast cell division [81,82]. Additional evidence indicates that in KRAS-mutant cancer cells, IRE1 regulates sensitivity to MEK inhibition [41] and to KRAS-mutant inhibition [44]. Furthermore, cleavage of IRE1 by Presenilin-1 yields proteolytic fragments that translocate from the ER to the nucleus [83]; and IRE1 plays a critical nonenzymatic role independent of XBP1 in facilitating VDJ recombination and B cell receptor generation in mouse pre-B cells [84].

The most significant translational implication of this study is that enzymatic IRE1 inhibition will likely not suffice to realize IRE1's full potential as a therapeutic target for cancer. Earlier work has shown that enzymatic inhibition of IRE1, either

at the kinase or RNase level, or XBP1 disruption, can be efficacious in models representing diverse cancers. We reveal that either enzymatic IRE1 inhibition or XBP1 knockdown fails to restrain malignant growth of certain cancer models that are nevertheless attenuated by IRE1 silencing. This discovery creates a strong conceptual rationale to explore novel disruptive strategies for IRE1 at the gene, mRNA, or protein level, e.g., gene therapy, antisense oligonucleotides, small-interfering RNAs, or protein-targeted degraders. Based on our findings, such interventions may be more broadly effective than IRE1 kinase or RNase inhibition and should be considered for cancer and perhaps other pathologies that may involve nonenzymatic IRE1 function. Specific biomarker strategies will likely be needed to better predict IRE1 dependency in order to guide optimal IRE1-disruptive therapy for individual patients.

## Materials and methods

### Cell culture and treatments

All cell lines were obtained or generated from an internal repository maintained at Genentech. AMO1 (vendor DSMZ, ref: ACC538), KMS27 (vendor JCRB, ref: JCRB1188), JJN3 (vendor DSMZ, ref: ACC541), L363 (vendor DSMZ, ref: ACC49), HCT116 (vendor ATCC, ref: CCL-247), OPM2 (vendor DSMZ, ref: ACC50), and KMS11 (vendor JCRB, ref: JCRB1179) cell lines were obtained from ATCC and authenticated by short tandem repeat (STR) profiles. U-2 OS WT Flp-In T-REx and U2OS IRE1-HaloTag cells were as described elsewhere [9]. Cells were tested to ensure mycoplasma free within 3 months of use and were freshly thawed every 6 months. All cell lines were cultured in RPMI1640 media supplemented with 10% (v/v) fetal bovine serum (FBS, Sigma), 2 mM glutaMAX (Gibco), 100 U/ml penicillin (Gibco), and 100 μg/ml streptomycin (Gibco) in a 5% $CO_2$ incubator at 37 °C.

Treatments included thapsigargin (Tg) (Sigma, 100 nM), Doxycycline (Dox) (Takara, 0.2 μg/ml), pan-caspase inhibitor Q-VD-Oph (QVD) (SelleckChem, 30 μM), JNK inhibitor Tanzisertib (JNKi) (Med Chem Express, various concentrations, described in Plantevin-Krenitsky and colleagues [54]), 5-azacitidine (5-aza) (Med Chem Express, various concentrations), CDK4/6 inhibitor Palbociclib (Med Chem Express, 1 μM, described in Toogood and colleagues [67]) and CDK1 inhibitor RO-3,306 (Med Chem Express, 9 μM, described in Sunada and colleagues [68]). IRE1 kinase inhibitors KI: KI1 G5758 (described in example 176 of patent application WO/2020/056089) or KI2 G9668 (described in Guttman and colleagues [50]) were used at 1–3 μM and were from Genentech. IRE1 KI1 was used in all experiments unless otherwise specified in the figure legend. IRE1 RI (MKC8866) was used at 1–3 μM and was synthesized based on Sanches and colleagues [52].

### shRNA knockdown of IRE1 or XBP1

AMO1, KMS27, JJN3, L363, HCT116, and OPM2 cell lines were transfected with a piggyBac Dox-inducible ODE vector containing a Puromycin resistance marker and simultaneously directing expression of three different short hairpin (sh) RNAs against *ERN1* (IRE1) (including 7, 8, and 9 together) or *XBP1* (including 4, 6, and 7 together); or one shRNA for non-targeting control (NTC) (see sequences below). The shRNAs were designed based on prior screening of multiple siRNAs by qPCR and immunoblot for effective knockdown and optimized for on-target selectivity based on two algorithms, namely, Target Scan (https://www.targetscan.org/vert_50/seedmatch.html) and BLAST. TransIT-X2 Dynamic Delivery system (Mirus) was used for transfection following the manufacturer's instructions. Transfected cells were cultured in Puromycin and surviving cells were single-cell cloned. Clones with the most complete knockdown of IRE1 or XBP1 were selected by immunoblot.

- IRE1 shRNA 7: 5′-AGAACAAGCTCAACTACTT-3′

- IRE1 shRNA 8: 5′-GCACGTGAATTGATAGAGA-3′

- IRE1 shRNA 9: 5′-GAGAAGATGATTGCGATGG-3′

- XBP1 shRNA 4: 5′-GGTATTGACTCTTCAGATT-3′

- XBP1 shRNA 6: 5′-GCAAGTGGTAGATTTAGAA-3′

- XBP1 shRNA 7: 5′-GCTGGAAGCCATTAATGAA-3′

- NTC shRNA: 5′ TAGATAAGCATTATAATTCCT

## CRISPR/Cas9 knock-in of kinase dead D688N IRE1

AMO1 shIRE1 cl.1 D688N cells were generated externally using CRISPR technology by Synthego. The guide RNA sequence used to introduce the D688N (GAC>AAC) point mutation in endogenous IRE1 was UGAGGAUGUUGUGUGG-CUUU. Pool was single-cell cloned and individual clones were characterized.

## Knockdown of IRE1 by siRNA or ASO

HCT116 cells were harvested using trypsin and suspended to a concentration of $2.5 \times 10^5$ cells/ml. Cells were transfected with NTC siRNA (5′-CTGGATGCCTCACGTGGTT-3′) or IRE1 targeting siRNA (5′-TGTTGTTTGTGTCAACGCT-3′), using RNAi lipofectamine (Invitrogen) and cultured on standard six-well plates until analysis. Alternatively, HCT116 cells were harvested using trypsin and suspended to a concentration of $1 \times 10^7$ cells/ml. $1 \times 10^6$ cells were electroporated with 10 µM of NTC ASO (5′-GACTATACGCGCAATA-3′) or IRE targeting ASO (5′-AATTGCCGGTCCTTCT-3′), using nucleofector reagent cocktail (Lonza), cuvettes (Lonza) and the nucleofector device (Amaxa) in the "HCT116" function. Four replicate electroporations were pooled for each condition and the cells were cultured on standard six-well plates until analysis.

## Transient cellular IRE1 rescue

HCT116 shIRE1 cl.9 cells were transfected with empty vector (EV), or with vector containing 2 kb of 5′ untranslated sequence of the IRE1 gene followed by shIRE1α resistant cDNA encoding either WT IRE1 or the K907A RNase-dead mutant of IRE1, using TransIT-X2 Dynamic delivery system (Mirus) on six-well plates.

## CRISPR/Cas9 knockout of Caspase-8

AMO1 shIRE1 cl.1 C8 KO cells were generated using CRISPR technology by co-transfecting a Cas9 containing plasmid, pRK-TK-Neo-Cas9, with a Caspase-8 (*CASP8*) targeting gRNA (5′-GCCTGGACTACATTCCGCAA-3′) cloned into a pLKO vector. AMO1 shIRE1 cl.1 cells were transfected using TransIT-X2 Dynamic Delivery system (Mirus), according to the manufacturer's protocol, and single-cell cloned. Clone with the most complete depletion of Caspase-8 was selected by immunoblot.

## Cell confluence by live-cell imaging

Cells were plated at $5 \times 10^3$ cells/well in ultra-low attachment (ULA) 96-well plates (7007, Corning) and centrifuged at $600 \times g$ for 5 min for spheroid formation. For HCT116 cells, $2 \times 10^3$ cells/well were seeded in a clear flat bottom 96-well plate (3,595, Corning). Treatments were added at the time of cell seeding and cells were cultured in complete RPMI media to a final volume of 200 µl/well. For the CM experiments, cells were instead seeded in CM collected from a confluent T-75 flask of either parental AMO1 (WT) or AMO1 shIRE1 cells treated with Dox for 24 h. The CM were filtered with a 0.2 µm filter to remove cells and stored at −80 °C until use. Cultures were maintained at 37 °C throughout the duration of the experiment. Cell confluence was tracked using a live-cell imaging IncuCyte Zoom system (Essen Bioscience). Picture frames were captured at 4-h intervals using a 4× objective and cell confluency (%) from the total well area as a function of time was calculated using the IncuCyte software.

## Cell viability and caspase activity assays

Cell viability, by Cell Titer Glo assay (Promega), and caspase 3/7 activity, by Caspase-Glo 3/7 assay (Promega), were assessed following the manufacturer's instructions. Luminescence was read on an Envision system (PerkinElmer).

## Subcutaneous xenograft growth and efficacy studies

Mice were housed in individually ventilated cages within animal rooms maintained on a 14:10-h, light:dark cycle. Animal rooms were temperature and humidity-controlled, between 20.0 and 26.1 °C and 30% to 70%, respectively, with 10–15 room air exchanges per hour. $10 \times 10^6$ AMO1 shIRE1 cl. 1, AMO1 shXBP1 cl. 1, KMS27 shIRE1 cl. 9, or KMS27 shXBP1 cl.13) were suspended in 1:1 HBSS and Matrigel (Corning) to a final volume of 100 μl and injected subcutaneously in the right flank of 6–8 weeks old female C.B-17 SCID mice. When tumors reached ~150–300 mm³, mice were randomized into the following treatment groups depending on the experiment: Vehicle (5% sucrose drinking water, changed once per week); Doxycycline (0.5 mg/ml Doxycycline in 5% sucrose drinking water, changed three times per week); IRE1 KI G5758 (250 mg/kg in 35% PEG400/55% water/10% DMSO, twice a day by oral gavage); or IRE1 RI (MKC8866, synthesized based on Sanches and colleagues [52]) (100 mg/kg in MCT formulation, twice a day by oral gavage). For tumor efficacy studies, at least 10 animals per group were treated and tumor growth was monitored until the last vehicle control animal reached humane endpoint (tumor volume reached or exceeded 2,000 mm³). Analyses and comparisons of tumor growth were performed using a package of customized functions in R (Version 4.1.0 (2021-05-18); R Foundation for Statistical Computing; Vienna, Austria, which integrates software from open-source packages (e.g., lme4, mgcv, gamm4, multcomp, settings, and plyr) and several packages from tidyverse (e.g., magrittr, dplyr, tidyr, and ggplot2) [85]. Briefly, as tumors generally exhibit exponential growth, tumor volumes were subjected to natural log transformation before analysis. All raw tumor volume measurements less than 8 mm³ were judged to reflect complete tumor absence and were converted to 8 mm³ prior to natural log transformation. Additionally, all raw tumor volume measurements less than 16 mm³ were considered miniscule tumors too small to be measured accurately and were converted 16 mm³ prior to natural log transformation. The same generalized additive mixed model (GAMM, described in Forrest and colleagues [85]) was then applied to fit the temporal profile of the log-transformed tumor volumes in all study groups with regression splines and automatically generated spline bases. This approach addresses both repeated measurements from the same study subjects and moderate dropouts before the end of the study. For pharmacodynamic studies, at least five animals per group were treated for 3.5 days. Six hours after the last oral dosing, animals were sacrificed, tumors collected, and protein or RNA extracted as described below.

## Ethics statement

Animals were maintained in accordance with the Guide for the Care and Use of Laboratory Animals (National Research Council 2011). Genentech is an AAALAC-accredited facility and all animal activities in this research study were conducted under protocols approved by the Genentech Institutional Animal Care and Use Committee. All protocols used (20–1476, 21–1878, 21-1878C, and 17-0869W) were reviewed and approved by the committee.

## Immunoblot analysis

Cells were lysed or tumor tissues were mechanically disrupted using Bead Ruptor Elite (Omni) in RIPA lysis buffer (20–188, Millipore) supplemented with Halt protease and phosphatase inhibitor cocktail (ThermoFisher Scientific) and kept on ice during 30 min. Lysates were cleared by centrifugation at 13,600 g for 15 min at 4 °C, and protein amount was analyzed by BCA protein assay (ThermoFisher Scientific). Protein was denatured by adding NuPAGE LDS buffer and DTT reducing buffer (Invitrogen) and incubating the samples at 95 °C during 5 min. Equal amounts of denatured protein were loaded in NuPAGE pre-cast gels (Invitrogen), fractioned by SDS-PAGE and electro-transferred to nitrocellulose membranes using

the iBLOT2 system (Invitrogen). Membranes were blocked in 5% nonfat milk solution for 1 h at room temperature and probed with the corresponding primary antibody at 1:1,000 dilution overnight at 4 °C. This was followed by incubation with the corresponding horseradish peroxidase-conjugated secondary antibody at 1:10,000 dilution during 1 h at room temperature. All secondary antibodies were from Jackson Laboratories. The primary antibodies are listed in Table 1. A housekeeping protein was added as loading control in every blot. Band area density quantification as a proxy of protein expression was performed using the image processing software ImageJ2 2.14.0/1.54f and normalized to housekeeping protein expression.

## RT-qPCR

RNA from tumor tissues or cells was extracted with RNeasy Plus kit (Qiagen) and quantified with a Nanodrop spectrophotometer (Thermo Fisher Scientific). Equal amounts of RNA were reverse transcribed and amplified using TaqMan RNA-to-C T 1-Step Kit kit on the ViiA 7 Real-Time PCR System (Applied Biosystems). TaqMan gene expression assay probes (Thermo Fisher Scientific) were used to measure the expression of *ERN1* (IRE1, Hs00176385_m1), *XBP1s* (Hs03929085_g1), *DGAT2* (Hs01045913_m1), *TP53* (P53, Hs01034249_m1), *CDKN1B* (P27, Hs00153277_m1), *CDKN1A* (P21, Hs00355782_m1), and *HPRT1* (Hs02800695_m1). The delta-delta Ct (ΔΔCt) values per gene were calculated by relating each individual $C_t$ value to its *HPRT1* housekeeping control, and then normalizing to the individual or averaged control condition. The relative quantification was calculated as $2^{-\Delta\Delta Ct}$.

## Cell cycle analysis by PI staining

Cells were grown in standard culture dishes, except for KMS11, which was grown in 3D using in ULA 96-well plates (7007, Corning). For cell cycle determination 1 million cells per condition were fixed in ice-cold 70% EtOH for at least 2 h at 4 °C. Cells were plated at $5 \times 10^3$ cells/well in ULA 96-well plates (7007, Corning) and centrifuged at 600$g$ for 5 min for spheroid formation. For the determination of cells in mitosis (M) inside the G2/M gate, cells were additionally stained with anti-pH3 conjugated with AF488 (06–570, Millipore) antibody diluted 1:500 in PBS-T (Phosphate Buffered Saline – 0.05% Tween-20) + 3% BSA (Bovine Serum Albumin), and incubated with rocking overnight at 4°C. After washing in PBS-T + 3% BSA, samples were treated with 100 µg/ml of RNase (Zymo research) for 15 min followed by incubation with 50 µg/ml propidium iodide (PI) for 20 min at room temperature. Samples were run on a FACSCelesta Cell Analyzer (BD). At least 50.000 singlets per condition were analyzed for lineal PI area staining for DNA content determination. When pH3 staining was performed, pH3 area was also determined. Cell cycle distribution in G0/G1, S, G2/M, and SubG1 phases was performed with FlowJo 10, using the Dean-Jett-Fox model and constraining the G2 CV to G1 CV. M population was calculated as % of cells in the G2/M gate. G2 cells alone were calculated as G2/M–M population.

## Cell division generation analysis by CFSE staining

AMO1 shIRE1 5–10 million cells/ml were washed twice in PBS (Phosphate Buffered Saline) to remove any serum and incubated with 1 µM carboxyfluorescein succinimidyl ester (CFSE, Invitrogen) for 10 min at room temperature. Labeling was stopped by adding 4–5 volumes of cold complete RPMI media and incubation on ice for 3 min. Cells were washed thrice in complete RPMI media and 0.2 µg/ml Dox was added to the treated conditions. Cells were then returned to the incubator. After 12 or 48 h, cells were harvested and 1 million cells were incubated with 1:100 LIVE/DEAD Fixable Near-IR Dead Cell Stain Kit (Life Technologies), according to the manufacturer's instructions. After washing, cells were fixed using FOXP3 Fix/Perm buffer set (BioLegend) and stored at 4 °C for a maximum of 7 days. At least 50,000 live singlets/condition were analyzed for CFSE area staining on a FACSCelesta Cell Analyzer (BD). Proliferation analysis and generation determination were performed using Modfit LT 6.0 software and the Cell Tracking Wizard Tool.

**Table 1. Primary antibodies used for immunoblot analysis.**

| Antigen | Reference | Species | Provider | Notes |
|---|---|---|---|---|
| IRE1 | 3,294 | Rabbit | Cell Signaling | |
| pIRE1 | | Rabbit | Genentech | Described in Harnoss and colleagues [30] |
| XBP1s | | Mouse | Genentech | Described in Harnoss and colleagues [30] |
| CD59 | ab133707 | Rabbit | Abcam | |
| Caspase-3 | 9,662 | Rabbit | Cell Signaling | |
| Cleaved Caspase-3 | 9,663 | Rabbit | Cell Signaling | |
| Caspase-8 | 9,746 | Mouse | Cell Signaling | |
| JNK | 9,252 | Rabbit | Cell Signaling | |
| pJNK | 4,668 | Rabbit | Cell Signaling | Phosphorylation at T183 and/or Y185 |
| P53 | 9,282 | Rabbit | Cell Signaling | |
| pP53 | 9,284 | Rabbit | Cell Signaling | Phosphorylation at S15 |
| P21 | 2,947 | Rabbit | Cell Signaling | |
| P27 | 2,552 | Rabbit | Cell Signaling | |
| 53 BP1 | 4,937 | Rabbit | Cell Signaling | |
| H2AX | 7,631 | Rabbit | Cell Signaling | |
| pH2AX | 05-636 | Mouse | Sigma | Phosphorylation at S139 |
| UHRF1 | 21402-1-AP | Rabbit | Proteintech | |
| H3 | ab1791 | Rabbit | Abcam | |
| H3K9me3 | A-4036–050 | Rabbit | Epigentek | |
| Cyclin A2 | 4,656 | Mouse | Cell Signaling | |
| Cyclin B1 | 4,138 | Rabbit | Cell Signaling | |
| Cyclin D2 | 3,741 | Rabbit | Cell Signaling | |
| Rb | 9,309 | Mouse | Cell Signaling | |
| pRb | 8,516 | Rabbit | Cell Signaling | |
| TK1 | 15691-1-AP | Rabbit | Proteintech | |
| CDC20 | 14,866 | Rabbit | Cell Signaling | |
| CDCA5 | 20,180 | Rabbit | Cell Signaling | |
| CDCA8 | 12465-1-AP | Rabbit | Proteintech | |
| Aurora Kinase B | 3,094 | Rabbit | Cell Signaling | |
| APC3 | 12,530 | Rabbit | Cell Signaling | |
| Geminin | 52,508 | Rabbit | Cell Signaling | |
| Securin | 13,445 | Rabbit | Cell Signaling | |
| KIF22 | ab75783 | Rabbit | Abcam | |
| CDK1 | 9,116 | Mouse | Cell Signaling | |
| pCDK1 | 4,539 | Rabbit | Cell Signaling | Phosphorylation at Y15 |
| CDK2 | 2,546 | Rabbit | Cell Signaling | |
| CDK6 | 133,331 | Rabbit | Cell Signaling | |
| E2F2 | NBP2–67,723 | Rabbit | Novus | |
| CDC25c | 4,688 | Rabbit | Cell Signaling | |
| pCDC25c | 9,527 | Rabbit | Cell Signaling | Phosphorylation at T48 |
| Lamin B1 | 13,435 | Rabbit | Cell signaling | |
| KDEL | NBP1–97,469 | Mouse | Novus | |
| REEP5 | 14643-1-AP | Rabbit | Proteintech | |
| B actin | 4,970 | Rabbit | Cell Signaling | Housekeeping control |
| GAPDH | 5,174 | Rabbit | Cell Signaling | Housekeeping control |

## Apoptosis detection by Annexin V and PI staining

Early and late apoptosis in AMO1 shIRE1 cl.1 and shXBP1 cl.1 cells treated with Dox was determined with the FITC Annexin V apoptosis detection kit with propidium iodide (PI) (BioLegend). Briefly, 1 million cells were stained following the manufacturer's instructions. At least 50,000 singlets were analyzed for Annexin V-AF488 and PI area staining on a FACSCelesta Cell Analyzer (BD). Percentage of cells in early (Annexin V⁺ PI⁻) or late (Annexin V⁺ PI⁺) apoptosis was determined with FlowJo 10.

## Bulk RNA sequencing (RNAseq)

**Sample preparation and sequencing data acquisition.** AMO1 shIRE1 cl.1, cl.3 or shXBP1 cl.1, cl.18 cells were harvested in biologic triplicates at 0, 24, 48, and 72 h after treatment with Doxycycline (Dox, 0.2 µg/ml). For bulk RNA sequencing (RNAseq), RNA from $1 \times 10^6$ cells was first extracted with RNeasy Plus kit (Qiagen), per the manufacturer's protocol. Total RNA was quantified with Qubit RNA HS Assay Kit (Thermo Fisher Scientific) and quality was assessed using RNA ScreenTape on 4,200 TapeStation (Agilent Technologies). For sequencing library generation, the Truseq Stranded mRNA kit (Illumina) was used with an input of 100 nanograms of total RNA. Libraries were quantified with Qubit dsDNA HS Assay Kit (Thermo Fisher Scientific) and the average library size was determined using D1000 ScreenTape on 4,200 TapeStation (Agilent Technologies). Libraries were pooled and sequenced on NovaSeq 6,000 (Illumina) to generate 30 million single-end 50-base pair reads for each sample.

**RNAseq data analysis.** The bulk RNA-seq experiment was processed using the HTSeqGenie pipeline in BioConductor [86]. First, reads with low nucleotide qualities (70% of bases with quality < 23) or matches to rRNA and adapter sequences were removed. The remaining reads were aligned to the human reference genome GRCh37/hg19 using GSNAP (v.2013-10-10-v2) [87], allowing a maximum of two mismatches per 75 base-sequence (parameters: "-M2 -n 10 -B 2 -i 1 -N 1 -w 200000 -E 1 –pairmax-rna=200000 –clip-overlap"). Transcript annotation was based on the Gencode genes database [88]. To quantify gene expression levels, the number of reads mapping unambiguously to the exons of each gene was calculated.

The resulting count matrix was analyzed in R (v.4.1.3; 03 November 2023) using the edgeR package (v.3.32.1) [89]. The count matrix was filtered to remove low expressed genes by keeping the features with at least 15 counts, considering library size differences, in more than the minimum number of samples of the experimental design. The resulting filtered count matrix was then log2-transformed and TMM-normalized with *edgeR::cpm(log=T)* and *edgeR::calcNormFactors(method= "TMM")*. For exploratory data analysis, the matrix was further filtered to the most variable genes selected using projection score [90] to focus on the major contributors to the variance in transcriptional state. This matrix was then scaled using *base:scale()* for posterior exploration. To understand the general clustering of the samples in transcriptional state, dimensionality reduction of the scaled matrix using principal component analysis was performed using PCAtools (https://github.com/kevinblighe/PCAtools; v.2.5.15).

Differential expression analyses were conducted using limma (v.3.46.0) [91]. To identify significantly interacting genes between gene-silencing backgrounds and time points, a polynomial spline model with three degrees of freedom was used for the design matrix. A large number of genes were significantly interacting and a more stringent false discovery rate adjusted *p*-value of $10^{-12}$ was used to perform exploratory data analysis. A heatmap of the resulting data matrix with rows annotated by GO terms [92] or KEGG pathways [93] was constructed for preliminary interpretation with the function*EmbolcallRNAseq::GOheatmap*(https://github.com/PechuanLab/EmbolcallRNAseq;v.0.0.1). Differential expression analysis using a more straightforward model for the interaction of the two gene silencing backgrounds at each time point was also fitted recapitulating similar results. Volcano plots were produced using the package Enhanced-Volcano (https://github.com/kevinblighe/ EnhancedVolcano; v.1.8.0). Gene set enrichment analysis was performed on the log2-transformed fold change given by the differential expression contrasts using the GSEA function from Cluster-Profiler [94] on the Hallmark Gene Set Collection [95] and the GO collection [92]. Pathways were considered to be significant if their

false-discovery-rate-adjusted $p$ value was less than 0.2. The list of cell cycle genes used to score cell cycle activity on the transcriptomic data can be found in Kowalczyk and colleagues, 2015 [96]. PROGENy [97] was used to score pathways of interest and obtain the TP53 pathway response genes.

## Proteomics analysis

**Sample preparation.** AMO1 shIRE1 cl.1, cl.3 or shXBP1 cl.1, cl.18 cells were harvested in biologic triplicates at 0 or 24 h after treatment with Doxycycline (Dox, 0.2 µg/ml). Samples were validated for IRE1 and XBP1 expression through immunoblotting. Protein from $1 \times 10^6$ cells was used for proteomics analysis. Cells were pelleted by centrifugation and washed 3× with PBS. Cells were lysed in 8 M urea, 50 mM Tris HCL, pH 8.0, 1X Roche Complete Protease Inhibitor. Samples were sonicated using an amplitude of 35–40% with 1-s pulses on and off for 20 s. Lysates were clarified by centrifugation at 14,000 g and supernatant was transferred to a new tube. 50 µg of each sample was digested overnight with trypsin. The samples were reduced for 1 h at RT in 12 mM DTT, followed by alkylation for 1 h at RT in 15 mM iodoacetamide. Trypsin was added to achieve an enzyme:substrate ratio of 1:20. Each sample was acidified to 0.3% TFA and subjected to solid-phase extraction (SPE) using Water HLB. The SPE process involved activating matrix with 4× additions of 500 µL of 70% acetonitrile; the matrix was equilibrated with 4× additions of 500 µL of 0.3% TFA; The sample was added to each well, and the wells were cleaned with 3 additions of 500 µL of 0.3% TFA. The samples were eluted using the solutions of 60% acetonitrile in 0.3% TFA—an initial volume of 200 µL followed by an additional 400 µL. The samples were frozen at −80 °C and lyophilized overnight.

**DIA mass spectrometry acquisition.** For DIA (data-independent acquisition) LC–MS/MS measurement, 1 µg per sample was analyzed by nano LC/MS with a Waters M-class HPLC system interfaced to a ThermoFisher Exploris 480. Peptides were loaded on a trapping column and eluted over a 75 µm analytical column at 350 nL/min; both columns were packed with XSelect CSH C18 resin (Waters); the trapping column contained a 5 µm particle, the analytical column contained a 2.4 µm particle. The column was heated to 55°C using a column heater (Sonation). A 90-min gradient was employed. The mass spectrometer was operated in data-independent mode. Sequentially, full scan MS data (60,000 FWHM resolution) from m/z 385–1,015 was followed by 61 × 10 m/z precursor isolation windows, another full scan from m/z 385–1,015 was followed by 61 × 10 m/z windows staggered by 5 m/z; products were acquired at 15,000 FWHM resolution. The maximum ion injection time was set to 50 ms for full MS and dynamic mode for products with 9 data points required across the peak; the NCE was set to 30.

**DIA data analysis.** DIA mass spectrometric data were analyzed using Spectronaut v18.3.230830.50606 [98] (Biognosys) through directDIA pipeline, offering an optimal spectral library-free pipeline. This indicates that DIA MS raw files were all directly searched against the reviewed Swiss-Prot Human protein database (October 2023, organism 9,606, 20,606 entries). For the identification of the total proteomic datasets, oxidation at methionine and acetylation at the protein N-terminal were set as variable modifications, while carbamidomethylation at cysteine was set as a fixed modification. Both peptide and protein-false discovery rate (FDR) (based on Q value) were controlled at 1%, and the data matrix was filtered according to Q value. All the other Spectronaut settings for identification and quantification were kept as default. This means that features such as "Inference Correction" was enabled, "Local Normalization" was used, and quantification was performed at the MS2 level using peak areas.

**Experimental design and statistical analysis for DIA data.** No statistical methods were used to pre-determine sample size. Sample sizes were $n = 3$ and 2 independent clones per condition. The sample injection order was completely randomized. Protein abundance analysis for DIA data was performed by R package MSstats v4.8.7 [99]. MSstats preprocessed the normalized peak intensities from Spectronaut, quantified protein abundance using top 200 features per protein, and performed differential abundance analysis. Log2 fold-change and standard error were estimated by linear mixed effect model for each protein. To test two-sided null hypothesis of no changes in abundance, the model-based test statistics were compared to the Student $t$ test distribution with the degree of freedom appropriate for each protein.

The resulting *p* values were adjusted to control the FDR with the method by Benjamini-Hochberg. Gene set enrichment analysis was performed on the ranked list. The list was obtained by multiplying the −log10 adjusted *p* value with the sign of log2 fold-change), provided by the different abundance analysis. This analysis was performed using the GSEA function from Cluster-Profiler for the GO terms or on the Hallmark Gene set Collection. Pathways with an adjusted *p* value less than 0.2 were considered statistically significant.

## Micronuclei and mitotic cell determination by microscopy

1 million AMO1 shIRE1 cl.1 or shXBP1 cl.1 cells previously treated were harvested in 1 ml of complete media and kept at room temperature. 300 µl of the cell suspension were transferred to a monolayer on a coatead cytoslide (Epredia) using the Cytospin 4 (Thermo Fisher Scientific Scientific). Sides were plunged into 4% PFA (from 16% PFA, Electron Microscopy Sciences) for fixation at room temperature during 30 min. Cell nuclei were counterstained with 1:2,500 Hoechst 33342 (Thermo Fisher Scientific) at room temperature during 20 min and slides were mounted using Prolong Gold antifade mountant (Thermo Fisher Scientific) and #1.5 coverslips. Images of 10 fields/condition selected at random were acquired with a 40× objective in an Echo Revolve microscope. Micronuclei and mitotic cells were manually counted using the Cell Counter plugin in the image processing software ImageJ2 2.14.0/1.54f and normalized to the control samples.

## Cytogenetic analysis

Cytogenetic analysis was performed by Karyotype Bioarray on at least 20 G-banded metaphase spreads per condition of the AMO1 shIRE1 cl.1 cell line after 24-h or 48-h treatment with Dox (0.2 µg/ml) or 48-h treatment with RI (3 µM). The experiment was repeated two independent times for untreated and Dox-treated conditions and results were combined. Briefly, 3 h before the end of either treatment, colcemid solution (Thermo Fisher Scientific) was added to the cell cultures at a final concentration of 0.5 µg/ml, and incubated at 37 °C, 5% $CO_2$. Cells were transferred to 15 ml centrifuge tubes, and spun at 300 g for 7 min, then resuspended in 0.075 KCL hypotonic solution. After 7 min at room temperature cells were spun at 300 g, 7 min, and resuspended in 3:1 methanol: acetic acid fixative. After 10 min the fixed cell suspensions were centrifuged as above. Cell pellets resuspended in 0.5 ml of fixative were used to create metaphase chromosome spreads by dropping a single drop of suspension onto each distilled water-soaked microscope slide. Slides were baked at 90 °C, 50 min, treated with 0.1% trypsin-EDTA, and stained with Wright's Giemsa stain in Gurr's Buffer at pH 6.8. Metaphase chromosome spreads were analyzed by brightfield microscopy at 1000× utilizing Leica Biosystems CytoVision karyotyping software, version 7.7. A chromosome count per metaphase plate was performed. Every spread displayed multiple chromosomal rearrangements with minor variation from spread to spread. Frequently displayed were one or more copies of derivatives of chromosomes 1, 2, 3, 4, 10, and 11; a duplication involving a portion of the chromosome 4 q-arm; additional chromatin of unknown origin on chromosomes 9, 10, 13, and 14; an interstitial deletion of a portion of the chromosome 5 q-arm; one or more copies of a chromosome 12 with a p- arm deletion; and 5 to 12 marker chromosomes. A marker is defined as "a structurally abnormal chromosome that cannot be unambiguously identified by conventional banding cytogenetics: https://doi.org/10.1159/000510090

## DNA quantification by flow cytometry

1 million AMO1 shIRE1 cl.1 or shXBP1 cl.1 previously treated cells were plated in 1 ml complete media. In order to exclude DNA loss from apoptotic cell death we treated all cells with QVD 30 µM for the duration of the experiment. For DNA staining, one drop of Hoechst 33342 Ready Flow (Invitrogen) was added and cells were returned to the incubator for 1 h at 37 °C. Cells were then harvested, washed in PBS, and incubated with 1:100 LIVE/DEAD Fixable Near-IR Dead Cell Stain Kit (Life Technologies), according to the manufacturer's instructions. At least 50,000 live singlets/condition were

analyzed for Hoechst area staining on a FACSCelesta Cell Analyzer (BD). DNA quantity was determined as the mean fluorescence intensity of Hoechst with FlowJo10and normalized to the control samples.

### DNA methylation by 5-mc ELISA

Genomic DNA (gDNA) of 5 million AMO1 shIRE1 or shXBP1 cells previously treated with or without Dox (0.2 µg/ml) for 24 h was isolated using the AllPrep DNA/RNA Mini Kit (Qiagen). gDNA was quantified using Nanodrop spectrophotometer (Thermo Fisher Scientific) and 2–4 µl of gDNA per sample (approximately 100 ng) were analyzed by MethylFlash Global DNA methylation (5-mC) ELISA Easy kit (Epigentek), following manufacturer's protocol. Absorbance was read on a SpectraMax M2 Microplate Reader (Molecular Devices) at 450 nm. Using the standard curve from the ELISA and the amount of gDNA/sample, percentage of methylated DNA (% 5-mC) is calculated and normalized to the control sample.

### Conventional electron microscopy

Conventional EM was used to visualize potential general trends in morphological changes during IRE1 or XBP1 depletion in AMO1 cells. The analysis focused on changes in nuclear morphology at 0, 24, or 48 h after Dox addition to the cell culture. Cells were fixed in ½ Karnovsky's fixative, postfixed in 2% osmium tetroxide, "en block" stained in 0.5% uranyl acetate and then dehydrated in an ascending series of ethanol and acetone. Embedding was in epoxy resin. After polymerization semithin sections were cut with a Leica UC7 ultramicrotome and placed on carbon-coated glass slides. Sections were imaged with a Zeiss Gemini 300 scanning electron microscope and a BSD1 backscattered electron detector. Cells with high-density heterochromatin (HC), euchromatin (EC), mitotic cells, and apoptotic cells were manually counted. Additional details can be found in the figure legends.

### Nuclei isolation for protein expression analysis

Nuclei from 10–15 million AMO1 cells were isolated using the Nuclei EZ Prep Nuclei Isolation kit (Sigma Aldrich), according to the manufacturer's protocol. The supernatant containing cytoplasmic components (WCL-N), as well as the nuclear extract (N) were subjected to protein extraction and immunoblotting as described in the above section. Whole cell lysate (WCL) from intact cells was used as a control.

### CRISPR/Cas9 tagging of endogenous IRE1 with HaloTag

Tagging of endogenous IRE1 in the c-terminus region with HaloTag in AMO1 shIRE1 cl.1 and HCT116 shIRE1 cl.9 cells was achieved using the same methodology and reagents described elsewhere [9]. Briefly, cells were co-transfected with a plasmid encoding Cas9 with the guide RNA and a homology-directed repair template plasmid targeted at C-terminus of *ERN1*, and single-cell cloned. Clone with similar IRE1 expression than its non-tagged counterparts and with full IRE1 tagging (as demonstrated by a clear band that ran slower than WT IRE1 by immunoblot) was selected.

### Immunofluorescence of IRE1

$1 \times 10^5$ U2OS or $0.3 \times 10^5$ HCT116 IRE1 wild type (WT) or IRE1-HaloTag cells were grown for 24 h on Millicell EZ Slide 4-well chamber slides (Millipore). 1h before fixation, Janelia 646 HaloTag ligand (1:1,000, Promega) was added directly to the media according to the manufacturer's instructions. For AMO1 cells, $1 \times 10^6$ IRE1 WT or IRE1-HaloTag cells were plated in 12-well dishes and Janelia 646 HaloTag ligand (1:1,000, Promega) was added for 1h. Cells were then washed in PBS, fixed in 4% PFA, and permeabilized with 0.1% Triton X-100. Next, cells were subjected to blocking solution (5% BSA, 0.5% gelatin, 0.05% Tween 20) during 15 min at room temperature, followed by overnight incubation with primary antibody rabbit anti-LaminB1 (1:100, ab16048, Abcam) at 4 °C. Samples were washed and incubated with the corresponding secondary antibody donkey anti-rabbit AF488 (Jackson ImmunoResearch Laboratories) at 1:500 in blocking

solution for 1 h at room temperature. After washing, cell nuclei were counterstained with 1:2,500 Hoechst 33342 (Thermo Fisher Scientific) at room temperature during 20 min, slides were detached from the chamber and mounted using Prolong Gold antifade mountant (Thermo Fisher Scientific) and #1.5 coverslips. For the AMO1 suspension cell line, cells were directly resuspended in 300 µl of PBS and added to a live-cell imaging 8-well chambered coverglass system (Cellvis) for direct imaging. Images were captured with a 63x objective in a Leica SP8 confocal microscope. Leica Application Suite X (LAS X) software was used to analyze colocalization between IRE1-HaloTag and LaminB1.

## Immuno-electron microscopy

AMO1 shIRE1 cl.1 HaloTag cells were grown to 50%–75% confluency in T25 flasks. They were fixed either with 4% para-formaldehyde (PFA) in 0.1 M Sorensen's phosphate buffer (PB), pH 7.4, for 15 min at room temperature, or with 2% PFA, 0.2% glutaraldehyde (GA) in 0.1 M PB, pH 7.4, for 2 h at room temperature. The fixative solutions were replaced by 0.6% PFA in 0.1 M PB, pH 7.4, for 13 days. After rinsing in PBS and blocking in 0.15% glycine in PBS, cells were scraped in 1% gelatin in PBS, pelleted at 800 RCF, and embedded in 12% gelatin. Small blocks of cell pellet were cryoprotected with 2.3 M sucrose, mounted on aluminum pins and frozen in liquid nitrogen [100]. To detect IRE1-HaloTag in ultrathin cryosections of these cell pellet blocks, the G9281 rabbit anti-HaloTag antibody (Promega) was used at a dilution in the range of 1:30–1:500 (1:500 is the working dilution for immunofluorescence).

## IRE1 gene expression across cell lines

Normalized IRE1 (*ERN1)* gene expression levels in the form of log2 transcripts per million + 1 across more than 1,500 cell lines were directly downloaded from the DepMap (https://depmap.org/portal/) database (data from 24Q2 release) and used to generate a scatter plot of the distribution of IRE1 expression.

## Statistics

All statistical analyses were performed using GraphPad Prism 10 (GraphPad Software). The statistical tests are noted in each figure and significance is indicated as follows: $*p < 0.05$; $**p < 0.005$; $***p < 0.001$; $****p < 0.00001$ consensus. A $p$ value > 0.05 was considered non-significant (ns).

## Schematics

All schematics were created with BioRender. Science Suite dba BioRender has granted a BioRender Industry Publication License in accordance with BioRender's Terms of Service and Industry License Terms. All schematics are accompanied by their specific citation in their respective figure legends.

## Supporting information

**S1 Fig. Growth of certain cancer cell lines requires IRE1 but not its enzymatic activity. (A)** Validation of IRE1 and XBP1s knockdown in AMO1 cells in vitro. Cells stably transfected with plasmids encoding Doxycycline (Dox)-inducible short hairpin RNAs (shRNAs) (3 distinct shRNAs in tandem) against either IRE1 or XBP1 were incubated for 48 h in the absence or presence of Dox at the indicated concentrations. Cells were then analyzed by Immunoblotting (IB). An independent clone shown for each gene (cl.1 for shIRE1 and cl.1 for shXBP1). **(B)** Validation of shRNA knockdown and enzymatic IRE1 inhibition in AMO1 cells. AMO1 shNTC, shIRE1 cl.1 and shXBP1 cl.1 cells were incubated for 48 h in the absence or presence of Dox (0.2 µg/ml), or IRE1 RI (3 µM), or IRE1 KI (KI2, 3 µM). Cells were analyzed by IB. Because XBP1s production requires both IRE1 kinase and RNase activity, its depletion confirms the inhibition of both functions. **(C)**

Same cells as in B were analyzed by RT-qPCR for mRNA levels of *XBP1s* and *DGAT2* (RIDD target). Data points are means of two technical replicates with error bars. Representative plot of three independent experiments shown. **(D)** Validation of shRNA knockdown and enzymatic IRE1 inhibition in AMO1 cells in vivo. C.B-17 SCID mice were implanted subcutaneously with $10 \times 10^6$ AMO1 shIRE1 cl.1 cells and allowed to form palpable tumors. Mice were then randomized into treatment groups (*n* = 5/group) and given either vehicle (5% sucrose) or Dox (0.5 mg/ml in 5% sucrose) *ad libitum*, or treated orally bidaily with vehicle, or IRE1 KI (250 mg/kg) or IRE1 RI (100 mg/kg) over 3.5 days. Tumors were collected 6h after the last oral dosing and analyzed by IB. **(E)** Mice were treated as in D and tumor samples were analyzed by RT-qPCR for XBP1s and DGAT2 mRNA levels as in C. Data points represent one biological replicate. Error bars represent SD. **(F)** Validation of XBP1 knockdown in vivo in AMO1 tumor xenografts. Tumors were collected at endpoint from the study depicted in Fig 1B. Tumor samples were analyzed by IB. **(G)** Tumor samples shown in F were analyzed as in E by RT-qPCR. Data points represent one biological replicate. Error bars represent SD. **(H)** Effect of IRE1 knockdown or KI or RI or KI+RI treatment on in vitro growth of AMO1 cells. Left: Cells were treated with Dox (0.2 μg/ml), or IRE1 KI1 (G5758) (1 μM) or KI2 (G9668) (1 μM), or IRE1 RI (1 μM), or both (1 μM each) for 72 h and analyzed by CellTiter-Glo. RLU = Relative Luminescence Units. Data points are the means of 10 technical replicates. Error bars indicate SEM. Representative plot of three independent experiments shown. Right: IB validation of IRE1 depletion or enzymatic inhibition. **(I)** Effect of IRE1 knockdown on in vitro proliferation of AMO1 shIRE1 and AMO1 shIRE1 kinase dead D688N mutant cells. Left: Cells were incubated in the absence (closed symbols) or presence (open symbols) of Dox (0.2 μg/ml). Proliferation, depicted as % confluence, was monitored by time-lapse microscopy in an Incucyte instrument. Data points are the means of 10 technical replicates. Representative plot of three independent experiments is shown. Right: IB validation of IRE1 depletion for AMO1 shIRE1 and AMO1 shIRE1 kinase dead D688N mutant cells. Cells were incubated with Dox (0.2 μg/ml) for the indicated time and analyzed by IB. **(J)** Effect of IRE1 or XBP1 knockdown in KMS27 cells in vitro. Cells stably transfected with plasmids encoding Dox-inducible shRNAs against NTC or IRE1 or XBP1 were incubated for the indicated time in the absence or presence of Dox (0.2 μg/ml) and analyzed by IB. An independent clone shown for each gene (cl.9 for shIRE1 and cl.13 for shXBP1). **(K)** Validation of knockdown and enzymatic IRE1 inhibition in vitro in KMS27 cells. Cells were incubated in the absence or presence of 0.2 μg/ml Dox or IRE1 KI (KI2, 3 μM) for 72 h and analyzed by IB. **(L)** Validation of knockdown and enzymatic IRE1 inhibition in vivo in KMS27 tumors. C.B-17 SCID mice implanted subcutaneously with $10 \times 10^6$ KMS27 shIRE1 cl.9 or shXBP1 cl.13 cells and allowed to form palpable tumors. Mice were then randomized into treatment groups (*n* = 4/group) and given either vehicle (5% sucrose) or Dox (0.5 mg/ml in 5% sucrose) *ad libitum*, or treated orally bidaily with either vehicle, IRE1 KI (250 mg/kg) or IRE1 RI (100 mg/kg), over 3.5 days. Six hours after the last oral dosing animals were sacrificed, tumors collected and analyzed by IB. **(M)** RT-qPCR analysis of XBP1s and DGAT2 mRNA in tumor samples described in L. Data points represent one biological replicate. Error bars represent SD. **(N)** Effect of IRE1 or XBP1 knockdown on in vitro proliferation of JJN3 cells. Cells were stably transfected with a plasmid encoding Dox-inducible shRNAs against NTC (black) IRE1 (red) or XBP1 (blue). Cells were incubated in the absence (closed symbols) or presence (open symbols) of Dox (0.2 μg/ml). Proliferation, depicted as % confluence, was monitored by time-lapse microscopy in an Incucyte instrument. Data points are means of 10 technical replicates. Representative plot of 3 independent experiments shown. An independent clone is shown for shIRE1 (cl.9). **(O)** Validation of IRE1 and XBP1 knockdown in JJN3 cells. Cells described in N were incubated with Dox (0.2 μg/ml) for 48 h and analyzed by IB. Thapsigargin (Tg, 100 nM) was added for 1 h to stimulate IRE1 activity. **(P)** Effect of IRE1 or XBP1 knockdown on in vitro proliferation of L363 cells. Cells were stably transfected with plasmids encoding Dox-inducible shRNAs against IRE1 (red) or XBP1 (blue). Two independent clones (circles or squares) for each gene knockdown were characterized. Cells were incubated in the absence or presence of Dox (0.2 μg/ml) and analyzed for proliferation as in N. Data points are means of 5 technical replicates. Representative plot of three independent experiments shown. **(Q)** Validation of knockdown in L363 cells. Cells were incubated with Dox (0.2 μg/ml) for the indicated time and analyzed by IB. **(R)** Effect of IRE1 or XBP1 knockdown on in vitro proliferation of HCT116 cells. Cells were stably transfected with plasmids encoding

Dox-inducible shRNAs against IRE1 (red) or XBP1 (blue). Cells were incubated in the absence (closed symbols) or presence (open symbols) of Dox (0.2 µg/ml) and analyzed for proliferation as in N. Data points are means of five technical replicates. Representative plot of three independent experiments shown. Independent clones are shown for shIRE1 (cl.9) and shXBP1 (cl.6). **(S)** Validation of knockdown in HCT116 cells. Cells described in R were incubated with Dox (0.2 µg/ml) for the indicated time and analyzed by IB. **(T)** Effect of IRE1 or XBP1 knockdown on in vitro proliferation of OPM2 cells. Cells were stably transfected with plasmids encoding Dox-inducible shRNAs against IRE1 or XBP1. Cells were incubated in the absence or presence of Dox (0.2 µg/ml) and analyzed for proliferation as in N. Data points are means of six technical replicates. Representative plot of three independent experiments shown. Independent clones are shown for shIRE1 (cl.6) and shXBP1 (cl.2). **(U)** Validation of knockdown in OPM2 cells. Cells described in T were incubated with Dox (0.2 µg/ml) for 72 h and analyzed by IB. **(V)** Open reading frame of IRE1 containing the canonical 22 exons. The position in the sequence of the strategies used to transcriptionally deplete IRE1 and their specific position are indicated by arrows for silencing RNA (siRNA) (green), shRNA (red), and Antisense Oligonucleotide (ASO) (orange). **(W)** Effect of IRE1 silencing by siRNA technology in HCT116 cells transfected with either NTC or IRE1 siRNAs after 48 h. Left: IB validation of IRE1 depletion. Right: In vitro viability was analyzed by CellTiter-Glo as in H. Data points are the means of 3 biological replicates. **(X)** Effect of IRE1 mRNA downregulation by ASO technology in HCT116 cells nucleofected with either NTC or IRE1 ASO after 48 h. Left: IB validation of IRE1 depletion. Right: In vitro viability was analyzed by CellTiter-Glo as in H. Data points are the means of three biological replicates. **(Y)** Effect of transient IRE1 reconstitution on HCT116 shIRE1 cl.9 cells. Left: HCT116 shIRE1 cells transfected with an empty vector (EV), Dox-resistant wild type (WT) IRE1, or RNase dead mutant (K907A) IRE1 were treated in the absence or presence of Dox (0.2 µg/ml) and analyzed by IB after 24 h. Right: In vitro viability was analyzed by CellTiter-Glo after 3 days of culture as in H. Results normalized to each non-Dox control condition and depicted as %RLU. Data points are the means of two biological replicates including three technical replicates each. Mann–Whitney $U$ statistical test. **$p < 0.005$. All raw data can be found in S1 Data.
(TIFF)

**S2 Fig. IRE1 depletion induces cell cycle arrest independently of apoptosis. (A)** AMO1 shIRE1 cl.1 or AMO1 shIRE1 kinase dead D688N mutant cl.1 cells were incubated with Dox (0.2 µg/ml) for the indicated time. Cells were stained with propidium iodide (PI) and analyzed by flow cytometry and cell frequencies by cell cycle phase were depicted as stacked bar graphs. Representative plot of 3 independent experiments shown. **(B)** Effect of QVD on AMO1 cell proliferation. AMO1 shIRE1 cl.1 cells were treated for 24 h with Dox (open symbols) (0.2 µg/ml) in the absence (circles) or presence (triangles) of QVD (30 µM) and analyzed for proliferation. Proliferation, depicted as % confluence, was monitored by time-lapse microscopy in an Incucyte instrument. Data points are mean of five technical replicates. Representative plot of 3 independent experiments shown. **(C)** AMO1 shIRE1 cl.1 cells were treated with JNK inhibitor Tanzisertib (JNKi) at the indicated concentrations for 48 h and analyzed for IRE1 and JNK pathway activation by IB. Representative blot of two independent experiments shown. **(D)** Effect of JNK inhibition on AMO1 cell proliferation. AMO1 shIRE1 cl.1 cells were incubated in the absence (closed symbols) or presence (open symbols) of Dox (0.2 µg/ml) or different concentrations of JNK inhibitor (JNKi, green). Proliferation was analyzed as in B. Data points are mean of five technical replicates. Representative plot of three independent experiments shown. **(E)** AMO1 shIRE1 cells were incubated with Dox (0.2 µg/ml) or different concentrations of JNK inhibitor (JNKi) for 48 h and analyzed as in A. **(F)** KMS27 shIRE1 cl.9 or shXBP1 cl.13 cells were incubated with Dox (0.2 µg/ml) for the indicated time and analyzed as in A. Representative plot of three independent experiments shown. **(G)** JJN3 shIRE1 cl.9 or shXBP1 cells were incubated with Dox (0.2 µg/ml) for the indicated time and analyzed as in A. Representative plot of three independent experiments shown. **(H)** L-363 shIRE1 or shXBP1 cells were incubated with Dox (0.2 µg/ml) for the indicated time. Two independent clones for IRE1 (cl.C and cl.D) or XBP1 (cl. B and cl.N) were used. Cells were analyzed as in A. Representative plot of three independent experiments shown. **(I)** HCT116 shIRE1 cl.9 or shXBP1 cl.6 cells were incubated with Dox (0.2 µg/ml) for the indicated time and analyzed as in A. Representative plot of three independent experiments

shown. **(J)** Validation of caspase activity and QVD-mediated inhibition in KMS27 cells. KMS27 shIRE1 cl.9 or shXBP1 cl.13 cells treated for the indicated time with Dox (0.2 µg/ml) and QVD (30 µM) and analyzed by IB. Representative plot of 3 independent experiments shown. **(K)** KMS27 shIRE1 cl.9 or shXBP1 cl.13 cells were treated as in J and analyzed for proliferation as in B. Data points are means of 6 technical replicates. Representative plot of three independent experiments shown. **(L)** Stacked bar graphs showing cell frequencies by cell cycle phase for KMS27 shIRE1 cl.9 or shXBP1 cl.13 cells treated as in J and analyzed as in A. Representative plot of three independent experiments shown. **(M)** JJN3 shIRE1 cl.9 cells were incubated with Dox (0.2 µg/ml) in the absence or presence of QVD (30 µM) and analyzed for confluence as in B. Data points are means of six technical replicates. Representative plot of three independent experiments shown. **(N)** HCT116 shIRE1 cells were incubated with Dox (0.2 µg/ml) in the absence or presence of QVD (30 µM) and analyzed for proliferation as in B. Data points are means of 10 technical replicates. Representative plot of three independent experiments shown. **(O)** Stacked bar graphs showing cell frequencies by cell cycle phase for OPM2 shIRE1 cl.6 or shXBP1 cl.2 cells incubated with Dox (0.2 µg/ml) for the indicated time. Cells were analyzed as in A. Representative plot of three independent experiments shown. **(P)** Stacked bar graphs showing cell frequencies by cell cycle phase for KMS11 shIRE1 or shXBP1 cl.8 cells incubated with Dox (0.2 µg/ml) for the indicated time in 3D culture conditions. Cells were analyzed as in A. Representative plot of 3 independent experiments shown. **(Q)** Flow cytometry histograms representing AMO1 shIRE1 cl.1 or shXBP1 cl.1 cells were for 72 h with Dox (0.2 µg/ml) or IRE1 KI or RI (3 µM) in the absence or presence of QVD (30 µM), stained with PI and FITC anti-Annexin V antibody. Cells were analyzed by flow cytometry for early (low PI, high FITC anti-Annexin V) or late apoptosis (high PI and FITC anti-Annexin V), as indicated in the quadrants. Representative plot of 3 independent experiments shown. **(R)** Frequency of cells shown in Q in early and late apoptosis. Results shown as fold change of apoptotic cells to the untreated controls. **(S)** IB validation of caspase-8 knockout in AMO1 shIRE1 cl.1 cells. Cells were subjected to CRISPR/CAS9-mediated disruption using guide RNA against caspase-8 (C8). Parental (C8 WT) or C8 knockout (C8 KO) cells were incubated for 72 h in the absence or presence of Dox (0.2 µg/ml) and analyzed by IB. **(T)** Caspase 3/7 activity of AMO1 shIRE1 cl.1 C8 WT or C8 KO cells incubated for 72 h with Dox (0.2 µg/ml). Cells were analyzed by Caspase-3/7 Glo assay. RLU = Relative Luminescence Units. Bars are means of three technical replicates ± SEM. Representative plot of 3 independent experiments shown. **(U)** AMO1 shIRE1 cl.1 C8 WT or C8 KO cells were incubated with Dox (0.2 µg/ml) and analyzed for confluence as in B. Data points are means of 10 technical replicates. Representative plot of 3 independent experiments shown. **(V)** Effect of conditioned media on the growth of AMO1 cells. Conditioned media were collected after 24 h of culture from naïve AMO1 cells [CM (WT)]. AMO1 shIRE1 cl.1 cells were then grown in the absence or presence of Dox (0.2 µg/ml) together with fresh media or CM (WT) and analyzed for confluence as in B. Data points are means of at least five technical replicates. Representative plot of three independent experiments shown. **(W)** CM were collected after 24 h of culture from AMO1 shIRE1 cl.1 cells grown in the absence or presence of Dox (0.2 µg/ml) [(CM (shIRE1) or CM (shIRE1+Dox), respectively]. AMO1 WT cells were then grown in the absence or presence of Dox, or CM (shIRE1) or CM (shIRE1+Dox) and analyzed for confluence as in B. Data points are means of at least six technical replicates. Representative plot of three independent experiments shown. All raw data can be found in S1 Data.
(TIFF)

**S3 Fig. IRE1 silencing downregulates cell cycle genes and engages TP53 and specific CDK inhibitors.** (A) RNA sequencing sample validation. AMO1 shIRE1 cl.1 or cl.3 cells and shXBP1 cl.1 and cl.18 cells were incubated for the indicated time with Dox (0.2 µg/ml) in triplicates and analyzed by IB. **(B)** Cells as in A were analyzed by RT-qPCR. Data points represent one biological replicate. Error bars represent SD. **(C)** Effect of IRE1 or XBP1 knockdown on mRNA expression of select XBP1s- and RIDD-target genes. Cells as in A were analyzed by bulk RNA sequencing (RNAseq) for mRNA expression of the XBP1s targets *Sec61A1* and *SYVN1* or the RIDD targets *DGAT2* and *BCAM*. **(D)** Row-clustered heatmap of the scaled mRNA expression by bulk RNAseq of genes involved in the S phase of the cell cycle from RNA sequencing of cells as in A, used to calculate the S phase score in Fig 3B. **(E)** Row-clustered heatmap of the scaled

mRNA expression by bulk RNAseq of genes involved in the G2 and M phases of the cell cycle from RNA sequencing of cells as in A, used to calculate the G2/M phase score in Fig 3C. **(F)** PROGENy pathway analysis of the effect of IRE1 or XBP1 knockdown on mRNA expression. PROGENy scaled pathway score heatmap depicting p53 as one of the most differentially altered pathways upon IRE1 versus XBP1s knockdown for samples depicted in Fig 3A. **(G)** Effect of IRE1 or XBP1 knockdown on mRNA expression of p53 target genes. Row-clustered heatmap depicting scaled mRNA expression by RNA sequencing of the top 100 TP53 pathway response genes for samples shown in Fig 3A, given the PROGENy model. **(H)** Effect of IRE1 or XBP1 knockdown on p53 cleavage. AMO1 shIRE1 cl.1 or shXBP1 cl.1 cells were incubated for the indicated time with Dox (0.2 µg/ml) and QVD (30 µM) and analyzed by IB. Representative blot of 3 independent experiments shown. **(I)** Effect of IRE1 or XBP1 knockdown on CDKN1B/p27 protein levels. Data depicted are from the proteomics analysis described in Fig 3F. Data points are mean ± SE for all biological and technical replicates normalized to $t = 0$ h for each cell line. **(J)** Effect of IRE1 knockdown on p53, p21, and p27 protein levels. AMO1 shIRE1 cl.1 and KMS27 shIRE1 cl.9 cells were incubated for the indicated time with Dox (0.2 µg/ml) and analyzed by IB. Representative blot of three independent experiments shown. **(K)** Effect of IRE1 or XBP1 knockdown on p27 mRNA levels. AMO1 shIRE1 cl.1 or shXBP1 cl.1 cells were incubated for the indicated time with Dox (0.2 µg/ml) and analyzed by RT-qPCR. Data points are one biological replicate normalized to its untreated counterpart. Representative plot of 3 independent experiments shown. **(L)** Effect of IRE1 or XBP1 knockdown on H2AX and phosphorylated (γ) H2AX (gH2AX). AMO1 shIRE1 cl.1 or shXBP1 cl.1 cells were incubated for the indicated time with Dox (0.2 µg/ml) and analyzed by IB. Representative blot of 3 independent experiments shown. **(M)** Effect of IRE1 or XBP1 knockdown on p53, p21, and 53 BP1 proteins. AMO1 shIRE1 cl.1 or shXBP1 cl.1 cells were incubated for the indicated time with Dox (0.2 µg/ml) and analyzed by IB. Representative blot of three independent experiments shown. All raw data can be found in S1 Data. (TIFF)

**S4 Fig. IRE1 depletion increases chromosome instability. (A)** Enrichment plots from the most significantly down-regulated Hallmark gene sets corresponding to proteins regulated upon IRE1 knockdown versus XBP1 knockdown after 24 h. Data from the proteomics experiment shown in Fig 3F was used for the Hallmark GSEA. **(B)** Enrichment plots as in A for GO GSEA. **(C)** Effect of IRE1 or XBP1 knockdown on the fraction of nuclear versus non-nuclear proteins impacted based on proteomics results from Fig 3F. **(D)** Additional examples of cells analyzed as depicted in Fig 4A. Blue circles indicate examples of cells with micronuclei, while red arrows indicate mitotic cells. Scale bar = 20 µm. **(E)** Effect of IRE1 or XBP1 knockdown on frequency of micronuclei and mitotic events. KMS27 shIRE1 cl.9 and shXBP1 cl.6 were cultured in the presence or absence of Dox (0.2 µg/ml) for 24 h, stained with Hoechst DNA dye, and analyzed by fluorescence microscopy. Images representative of at least 10 fields examined per condition of three independent experiments. Blue circles indicate examples of cells with micronuclei, while red arrows indicate mitotic cells. Scale bar = 20 µm. **(F)** Quantification of annotated events in E normalized per total amount of cells per field. Ten fields per condition with at least 100 cells were counted for $n = 3$ independent experiments. Mean ± SEM of the fold change to untreated controls for each cell line. Mann–Whitney $U$ statistical test. **(G)** Quantification of mitotic cells exemplified in E per total amount of cells per field. Ten fields per condition with at least 100 cells were counted for three independent experiments. Mean ± SEM of the fold change to untreated controls for each cell line. Mann–Whitney $U$ statistical test. $*p < 0.05$; $**p < 0.005$; $***p < 0.001$; $****p < 0.00001$ consensus. A $p$ value $> 0.05$ was considered non-significant (ns). All raw data can be found in S1 Data. (TIFF)

**S5 Fig. IRE1 depletion decreases heterochromatin, DNA, and H3K9me3 methylation, and UHRF1. (A)** Additional examples of cells analyzed as depicted in Fig 5A and 5B. **(B)** Examples of cells from an independent unbiased EM analysis performed at a different institution (Utrecht). AMO1 shIRE1 cl.1 cells were incubated for the indicated time with Dox (0.2 µg/ml). Scale bars are 1 µm. **(C)** AMO1 shIRE1 cl.1 cells were incubated for 24 h in the absence or presence of Dox (0.2 µg/ml) with the indicated concentration of 5-aza and analyzed by IB. The quantification for these IBs is depicted in Fig 5D. **(D)** Effect

of XBP1 knockdown and 5-aza treatment on proliferation. AMO1 shXBP1 cl.1 cells were incubated for the indicated time in the absence (closed symbols) or presence (open symbols) of Dox (0.2 µg/ml), without (circles) or with (triangles) 5-aza (1 µM). Proliferation, depicted as % confluence, was monitored by time-lapse microscopy in an Incucyte instrument. Data points are means of five technical replicates. Representative plot of three independent experiments shown. **(E)** Effect of IRE1 or XBP1 knockdown on mRNA expression of heterochromatin formation and DNA methylation genes. Row-clustered heatmap depicting scaled mRNA expression by RNA sequencing of genes composing GO terms *Heterochromatin Formation*, *Regulation of DNA Methylation-dependent Heterochromatin Formation*, and *H3K9me3 Modified Histone Binding* for samples shown in Fig 3A. **(F)** Effect of IRE1 or XBP1 knockdown in vivo on UHRF1, p53, p21, and p27 protein levels. AMO1 tumor xenografts as depicted in S1D Fig were analyzed by IB. Note that the IRE1 and GAPDH blots from S1D Fig are duplicated here for direct reference. **(G)** KMS27 tumor xenografts as depicted in S1M Fig were analyzed by IB. Note that the IRE1 and GAPDH blots from S1M Fig are duplicated here for direct reference. All raw data can be found in S1 Data.
(TIFF)

**S6 Fig. IRE1 depletion downregulates multiple cell cycle proteins in synchronized cells. (A)** AMO1 shNTC, shIRE1 cl.1, shIRE1 cl.1 IRE1 D688N kinase dead cl.1, and shXBP1 cl.1 cells were incubated for 48 h in the absence or presence of Dox (0.2 µg/ml), or IRE1 KI or RI (3 µM) and analyzed by IB. **(B)** AMO1 shIRE1 cl.1 cells were incubated for the indicated time in the presence of Dox (0.2 µg/ml) or IRE1 RI (3 µM) and analyzed by IB, together with AMO1 shIRE1 cl.1 IRE1 kinase dead D688N cl.1 and shXBP1 cl.1 cells. **(C)** Graphs depicting cell frequencies from Fig 6A for each cell cycle phase for G1-synchronized AMO1 shIRE1 cl.1 cells in the presence (open symbols) or absence (closed symbols) of Dox (0.2 µg/ml). Each dot represents one biological replicate. Representative plot of three independent experiments shown. **(D)** Graphs depicting cell frequencies from Fig 6C for each cell cycle phase for late G2-synchronized AMO1 shIRE1 cl.1 cells in the presence (open symbols) or absence (closed symbols) of Dox (0.2 µg/ml). Each point depicts one biological replicate. Representative plot of three independent experiments shown. All raw data can be found in S1 Data.
(TIFF)

**S7 Fig. Endogenous IRE1 protein can localize to the nuclear envelope. (A)** AMO1 shIRE1 cl.1 cells were incubated in the presence or absence of Dox (0.2 µg/ml) for 24 h, subjected to subcellular fractionation, and analyzed by IB. WCL: whole-cell lysate. WCL-Nuc: Whole-cell lysate lacking the nuclear fraction. Nuc: nuclear fraction. The nuclear protein Lamin B1 and the ER markers KDEL and REEP5 were used to confirm the purity of the nuclear fraction. Representative blot of 2 independent experiments shown. **(B)** Additional controls of the experiment described in Fig 7B performed in both U2OS IRE1 HaloTag and U2OS wild type IRE1 (IRE1 WT) cells. Scale bar = 2 µm. Representative image of 2 independent experiments shown. **(C)** AMO1 shIRE1 cl.1 with wild type IRE1 (WT IRE1) or AMO1 shIRE1 cl.1 with endogenously tagged IRE1 (IRE1 HaloTag) cells were incubated for the indicated time in the presence of Dox (0.2 µg/ml) and analyzed by IB. Note that Halo-tagged IRE1 runs slower that WT IRE1 due to the increase in size produced by the HaloTag. Representative blot of three independent experiments shown. **(D)** and **(E)** Detection of endogenous Halo-tagged IRE1 and non-tagged Lamin B1 by confocal microscopy in AMO1 **(D)** and HCT116 **(E)** in both IRE1 Halo-tagged and WT cells. Cells were cultured in the presence of Janelia 646 HaloTag ligand that detects IRE1-HaloTag (red), fixed, stained with anti-Lamin-B1 (green) antibody, and analyzed by confocal microscopy. Individual and merge fields of IRE1 (red), Lamin B1 (green), and nucleus (blue) are shown. Scale bar = 5 µm. Pixel size = 48 nm for all images. Representative images of 2 independent experiments shown. Of note, AMO1 cells present nuclear autofluorescence in the far-red channel (HaloTag 646 channel) that doesn't correspond to specific IRE1 staining as shown in the controls. **(F)** and **(G)** Left: Merge (left) and Close up (right) of the nuclear envelope for IRE1 and LaminB1 staining in AMO1 **(F)** and in HCT116 **(G)**. Scale bar = 5 µm (merge) and 2 µm (close up). Pixel size = 48 nm for all images. **(H)** Additional examples of the staining described in Fig 7C: (a) additional example of IRE1-HaloTag detection in AMO1 cells. Arrowheads and arrows indicate gold particles seen in association with ER membranes or nuclear envelope membranes, respectively. N = nucleus. (b) Control staining

for protein-A-Gold[10] without anti-HaloTag antibody. (c) and (d) Control staining for AMO1 shIRE1 cl.1 cells after 48 h of incubation with Dox (0.2 µg/ml) to deplete IRE1. **(I)** Scatter plot showing normalized IRE1 (*ERN1*) gene expression levels in the form of log2 transcripts per million (TPM) + 1 across cell lines contained in the DepMap database. Cell lines used in this study are marked in red. Error bar represents SD.
(TIFF)

**S1 Raw Images.  Original raw immunoblot images used for figures.** Each raw blot is labeled to annotate the loading order, experimental sample identity, protein blotted for and the figure that was generated from that original image. Lanes not included in the final figures are marked with an "X" above the lane label.
(TIFF)

**S1 Data.  Original raw data used to generate graphs in Figs 1–6 and S1–S6 Fig.**
(XLSX)

## Acknowledgments

We thank Vladislav Belyy for advice regarding HaloTag technology, and Suzie Scales and Meredith Sagolla for coordinating EM studies. We thank Lauren Gutgesell for help with cell line characterization and Ofer Guttman, Alan Gutierrez, and Bruno Alicke for assistance with in vivo models. We thank René Scriwanek for help with figure layout and Robert Piskol for help with RNAseq data curation in GEO database. The EM infrastructure of UMC Utrecht used in this work is part of the National Roadmap for Large-Scale Research Infrastructure (NEMI), financed by the Dutch Research Council (NWO): project number 184.034.014.

## Author contributions

**Conceptualization:** Iratxe Zuazo-Gaztelu, Avi Ashkenazi.

**Data curation:** Iratxe Zuazo-Gaztelu, David Lawrence, Ioanna Oikonomidi, Scot Marsters, Ximo Pechuan-Jorge, Catarina J. Gaspar, David Kan, Ehud Segal, Kevin Clark, Maureen Beresini, Meena Choi, Wendy Sandoval, Mike Reichelt, Pekka Kujala, Suzanne van Dijk.

**Formal analysis:** Iratxe Zuazo-Gaztelu, David Lawrence, Ioanna Oikonomidi, Scot Marsters, Ximo Pechuan-Jorge, Catarina J. Gaspar, David Kan, Ehud Segal, Kevin Clark, Maureen Beresini, Meena Choi, Wendy Sandoval, Mike Reichelt, Pekka Kujala, Suzanne van Dijk, Judith Klumperman.

**Methodology:** Iratxe Zuazo-Gaztelu, David Kan, Ehud Segal, Kevin Clark, Mike Reichelt, David C. DeWitt, Pekka Kujala, Suzanne van Dijk, Judith Klumperman.

**Resources:** Scot Marsters, Marie-Gabrielle Braun.

**Software:** Ximo Pechuan-Jorge, Meena Choi, David C. DeWitt.

**Supervision:** Joachim Rudolph, Zora Modrusan, Judith Klumperman, Avi Ashkenazi.

**Visualization:** David C. DeWitt.

**Writing – original draft:** Iratxe Zuazo-Gaztelu, Avi Ashkenazi.

**Writing – review & editing:** Iratxe Zuazo-Gaztelu, Zora Modrusan, Meena Choi, Mike Reichelt, Pekka Kujala, Judith Klumperman, Avi Ashkenazi.

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
