## [Editor Report · Decision Letter 0]

7 May 2024

Dear Dr Ashkenazi,

Thank you for submitting your manuscript entitled "A nonenzymatic dependency on inositol-requiring enzyme 1 controls cancer cell cycle progression and tumor growth" for consideration as a Research Article by PLOS Biology.

Your manuscript has now been evaluated by the PLOS Biology editorial staff as well as by an academic editor with relevant expertise and I am writing to let you know that we would like to send your submission out for external peer review.

Once your full submission is complete, your paper will undergo a series of checks in preparation for peer review. After your manuscript has passed the checks it will be sent out for review. To provide the metadata for your submission, please Login to Editorial Manager (https://www.editorialmanager.com/pbiology) within two working days, i.e. by May 09 2024 11:59PM.

Kind regards,

Ines

--

Ines Alvarez-Garcia, PhD

Senior Editor

PLOS Biology

---

## [Decision Letter · Decision Letter 1]

24 Jul 2024

Dear Dr Ashkenazi,

Thank you for your patience while your manuscript entitled "A nonenzymatic dependency on inositol-requiring enzyme 1 controls cancer cell cycle progression and tumor growth" was peer-reviewed at PLOS Biology. Please also accept again my apologies for the delay in sending you our decision. The manuscript has now been evaluated by the PLOS Biology editors, an Academic Editor with relevant expertise, and by three independent reviewers.

The reviews are attached below. As you will see, the reviewers find the conclusions interesting and worth pursuing, but they also raise several concerns that should be addressed before we consider the manuscript for publication. Reviewer 1 notes that all the findings rely on shRNA knockdowns that have potential for off-target effects, thus this reviewer thinks that a rescue experiment using a shRNA-resistant IRE1 construct should be performed to validate that the main downstream effects are due to the absence of IRE1. This reviewer also finds the methods a bit confusing, and raises other issues that should be addressed. Reviewer 2 suggests several experiments to confirm the findings, such as analysing the role of the IRE1-TRAF2-JNK pathway in cancer development, and several missing controls, among other issues. Reviewer 3 thinks you should check if the inhibitors used have indirect effects in the cell culture models, and discuss if IRE1 could be overexpressed in cancer cells to gain entry into the nucleus, acquiring new functions in cancer cells.

In light of the reviews and after discussion with the Academic Editor, we would like to invite you to revise the work to thoroughly address the reviewers' reports. Given the extent of revision needed, we cannot make a decision about publication until we have seen the revised manuscript and your response to the reviewers' comments. Your revised manuscript is likely to be sent for further evaluation by all or a subset of the reviewers.

**IMPORTANT - SUBMITTING YOUR REVISION**

3. Resubmission Checklist

a) *PLOS Data Policy*

b) *Published Peer Review*

Sincerely,

Ines

--

Ines Alvarez-Garcia, PhD

Senior Editor

PLOS Biology

Reviewers' Comments

Rev. 1:

This study by the Askenazi group reports the discovery that some cancer cells are dependent on IRE1 but not on its enzymatic activities that mediate XBP1 activation or RIDD. In such cells, the absence of IRE1 results in cell cycle arrest, p53 activation, DNA damage, and chromosome instability. The main implications of this finding are two-fold. First, IRE1 has a clear non-enzymatic role in regulating cell cycle progression and DNA damage. Second, this alternative role is a potential therapeutic target in some cancers where simply targeting IRE1's enzymatic activities would not work.

The conclusion, if validated, would be a very important advance even if the molecular mechanisms of how IRE1 links to DNA damage and cell cycle progression are not clear yet. Thus, with a suitable validation of the non-enzymatic function of IRE1, I am strongly supportive of publication.

Specific suggestions:

1) All of the downstream consequences of the putative non-enzymatic IRE1 function are well documented and clear. I therefore have no issues with any of this, although I admit that many of those areas are not my area of expertise. I am however concerned that all of the findings rely on shRNA knockdowns that have the potential for off-target effects. Thus, it seems critical to verify that the main downstream effects are due to the absence of IRE1. This is best accomplished by rescue experiments in which an shRNA-resistant IRE1 construct is used. In the absence of such a rescue, one would be concerned that the effects are due to perturbing some other gene product(s) whose absence is the basis of the observed phenotypes. I am not suggesting that everything be repeated with a rescue, but it does seem important to be certain that at least the key phenotypes are due to IRE1.

2) The methods are slightly confusing to me. Does the BAC have all three shRNAs for IRE1, or are there three BACs each with a different shRNA? If the latter, did the authors sequence the shRNAs in the selected cell lines to know which one is actually being used? Right now, I cannot work out from the Methods which shRNA is actually responsible for the effects observed. Please clarify.

3) The fact that some IRE1 can reside in the nuclear fraction is neither interesting nor illuminating. The ER is continuous with the outer NE, so unless a membrane protein has a preference for tubular ER and is unstable in flat membranes, it will be found in both the bulk ER and outer NE. The only situation where seeing nuclear IRE1 would be of interest is if it is in the inner NE and hence facing the nucleoplasm. If the authors are not able to show this, then I would eliminate the nuclear localization observation (or at least not make such a big deal of it).

Rev. 2:

This study investigated the role of endoplasmic-reticulum resident inositol-requiring enzyme 1α (IRE1) in cancer cell progression independent of its enzymatic activity. Knockdown of IRE1 expression attenuated cancer cell cycle and tumor growth, unlike enzymatic IRE1 inhibition or XBP1 knockdown. Mechanistically, IRE1 knockdown increased TP53 and CDKN1A/p21 expression, promoted DNA and H3 methylation, chromosome instability, and heterochromatin formation. Additionally, IRE1 was found to localize to the nuclear envelope. It is concluded that IRE1a regulates cancer cell progression independent of its enzymatic activation.

General Comments:

The study suggests a nonenzymatic role for IRE1 in cancer cell development, but there are significant issues with over-conclusions and experimental reliability.

Specific Comments:

1. IRE1-specific Inhibitors: The study uses IRE1-specific inhibitors, but these can affect other pathways. Constructing vectors for IRE1 enzyme inactivation would better verify its function.

2. Contradictory Results with XBP1: Figures 2B and 2C show that XBP1 silencing or enzymatic IRE1 inhibition does not affect the cell cycle, contradicting previous findings that XBP1 regulates cell cycle genes. The authors should further investigate this discrepancy.

3. Mechanism of IRE1 Silencing: The study shows IRE1 silencing downregulates many cell cycle genes independently of XBP1. The regulatory mechanism needs further clarification, including results from IRE1 enzyme inactivation.

4. Expression of p53 and p21: Figure 3H shows decreased p53 and increased p21 with shIRE1. The authors need to explain this unexpected result.

5. IRE1-TRAF2-JNK Pathway: The study should examine the IRE1-TRAF2-JNK pathway's role in cancer development to support the hypothesis.

6. Micronuclei Frequency: Figure 4B shows significant micronuclei increase with IRE1 silencing, but Figure 4A does not. This experiment should be repeated in other cancer cell lines.

7. Statistical Significance: Figures 4B, 4C, 4D, and 4E lack p-values and should include them for clarity.

8. Choice of UHRF1 Gene: The rationale for choosing UHRF1 over other genes involved in DNA methylation and chromatin modification should be explained. The RNA-seq results should also include DNA demethylation genes.

9. Cyclin Protein Expression: To prove IRE1's independent regulation of cyclin protein expression, add a kinase mutant group and ShRNA-XBP1 group in Figures 6 and S6.

10. Immunofluorescence Clarity: Figure 7B immunofluorescence results are unclear and lack a scale. The experiment should be repeated in other cancer cell lines.

11. Nuclear Localization of IRE1: The study claims IRE1 localizes in the nucleus, but the mechanism and its relation to enzyme activity and cell cycle regulation need clarification.

Rev. 3:

Zuazo-Gaztelu et al. describe a non-enzymatic function of IRE1 in regulating the cell cycle in certain cancer cells. IRE1 is an endoplasmic reticulum (ER) localized transmembrane kinase and RNase. Upon ER stress, IRE1 activates the XBP1 transcription factor, which upregulates many genes responsible for mitigating ER stress. IRE1 can also directly cleave certain mRNAs and miRNAs, which are implicated in reducing ER load and inducing cell death. In this manuscript, the authors have devised an elegant approach to deplete IRE1 using acutely shRNA in various cancer cell lines. They found that acute depletion of IRE1 resulted in inhibition of cell cycle progression in cancer cells in both in vitro and in vivo. By performing RNA seq, proteomics, and imaging, the authors observed the activation of the p53 pathway, DNA damage, and chromosome instability in some cancer cells. Therefore, the authors conclude that IRE1 functions as a scaffold to regulate cell cycle progression via unknown factors. Overall, this manuscript provides a new function of IRE1 and opens a new way to target IRE1 for therapeutic benefits. I suggest that the authors address the following concerns.

1. The use of IRE1 kinase and RNase inhibitors suggests IRE1's non-enzymatic function. Inhibitors could have indirect effects. The authors support their core findings by complementing IRE1 kinase and RNase mutants into IRE1-depleted cells. This can be done only in cell culture models and does not necessarily need to be shown with animal models.

2. It is difficult to rationalize that a low-abundant protein, IRE1 (~416 molecules/HeLa), functions as a scaffold protein that regulates the cell cycle (PMID: 24487582). Also, most endogenous IRE1 is bound to the Sec61 translocon in cells (PMID: 25993558). It is possible that IRE1 is overexpressed in cancer cells to gain entry into the nucleus, thus acquiring new functions in cancer cells. The authors should discuss this.

Minor concerns:

1. The resolution of the main figures is very poor.

2. The scale bar is missing in Figures 4A and 7B.

3. It is unclear why XBP1s is noticeably detected only upon Tg treatment in Fig. S1O, whereas it is detected without ER stress in other figures.

---

## [Decision Letter · Decision Letter 2]

23 Dec 2024

Dear Dr Ashkenazi,

Thank you for your patience while we considered your revised manuscript entitled "A nonenzymatic dependency on inositol-requiring enzyme 1 controls cancer cell cycle progression and tumor growth" for publication as a Research Article at PLOS Biology. This revised version of your manuscript has been evaluated by the PLOS Biology editors, the Academic Editor and by two of the original reviewers.

Based on the reviews (attached below) and on our Academic Editor's assessment of your revision, we are likely to accept this manuscript for publication, provided you satisfactorily address the data and other policy-related requests stated below. We discussed the remaining experiments requested by Reviewer 2 with the Academic Editor and, while we agree it would be nice to have these data, this would also require a lot of additional experiments. Thus, given that the other two reviewers are satisfied and that these were not requested in the previous round of review, we will not require these results for publication.

We expect to receive your revised manuscript by January 13.

*Published Peer Review History*

*Press*

Sincerely,

Ines

--

Ines Alvarez-Garcia, PhD

Senior Editor

PLOS Biology

Fig. 1B-I; Fig. 2A-D, F; Fig. 3A-G, J; Fig. 4B-E; Fig. 5B-F; Fig. 6A, C; Fig. S1C, E, G-I; Fig. S2A, B, D-H; Fig. S3B, C; Fig. S4A, B and Fig. S5D, E

***In addition, please make the data you have deposited in the ProteomeXChange Consortium publicly available at this stage.

CODE POLICY

We require the original, uncropped and minimally adjusted images supporting all blot and gel results reported in an article's figures or Supporting Information files. We will require these files before a manuscript can be accepted so please prepare and upload them now. Please carefully read our guidelines for how to prepare and upload this data: https://journals.plos.org/plosbiology/s/figures#loc-blot-and-gel-reporting-requirements

We do require the raw gels shown in the following figures:

Fig. 2E; Fig. 3H, I; Fig. 5G; Fig. 6B, D; Fig. 7; Fig. S1A, B, D, F, H-K; Fig. S2C; Fig. S3A; Fig. S4C, F, G; Fig. S6A and Fig. S7A, C

Reviewers' comments

Rev. 1:

The authors have performed the rescue experiment that I felt was important and this has strengthened the primary conclusions of the study. I am happy to support publication.

Rev. 2:

This study provides substantial data exploring the role of endoplasmic reticulum-resident inositol-requiring enzyme 1α (IRE1) in cancer cell progression, independent of its enzymatic activation. However, for data related to the cell cycle and genome stability, such as those shown in Figures 3, 4, and 6, the authors only performed experiments using shIRE1 and shXBP1. I believe that adding experiments with a kinase-dead IRE1 mutant, similar to those in Figure 1, would not only strengthen the argument, but also required to be published in Plos Biol. Without this, it remains unclear whether the regulation of the cell cycle and genome stability is independent of IRE1's enzymatic activity.

---

## [Editor Report · Decision Letter 3]

26 Feb 2025

Dear Dr Ashkenazi,

Thank you for the submission of your revised Research Article entitled "A nonenzymatic dependency on inositol-requiring enzyme 1 controls cancer cell cycle progression and tumor growth" for publication in PLOS Biology. On behalf of my colleagues and the Academic Editor, Ursula Jakob, I am delighted to let you know that we can in principle accept your manuscript for publication, provided you address any remaining formatting and reporting issues. These will be detailed in an email you should receive within 2-3 business days from our colleagues in the journal operations team; no action is required from you until then. Please note that we will not be able to formally accept your manuscript and schedule it for publication until you have completed any requested changes.

PRESS

Sincerely, 

Ines

--

Ines Alvarez-Garcia, PhD

Senior Editor

PLOS Biology
